# AKT1 phosphorylation of cytoplasmic ME2 induces a metabolic switch to glycolysis for tumorigenesis

Taiqi Chen[1,2,3,7], Siyi Xie[2,3,7], Jie Cheng[2,3], Qiao Zhao [4], Hong Wu [3,5] ✉, Peng Jiang [1,2,3] ✉ & Wenjing Du [6] ✉

Many types of tumors feature aerobic glycolysis for meeting their increased energetic and biosynthetic demands. However, it remains still unclear how this glycolytic phenomenon is achieved and coordinated with other metabolic pathways in tumor cells in response to growth stimuli. Here we report that activation of AKT1 induces a metabolic switch to glycolysis from the mitochondrial metabolism via phosphorylation of cytoplasmic malic enzyme 2 (ME2), named ME2fl (fl means full length), favoring an enhanced glycolytic phenotype. Mechanistically, in the cytoplasm, AKT1 phosphorylates ME2fl at serine 9 in the mitochondrial localization signal peptide at the N-terminus, preventing its mitochondrial translocation. Unlike mitochondrial ME2, which accounts for adjusting the tricarboxylic acid (TCA) cycle, ME2fl functions as a scaffold that brings together the key glycolytic enzymes phosphofructokinase (PFKL), glyceraldehyde-3-phosphate dehydrogenase (GAPDH) and pyruvate kinase M2 (PKM2), as well as Lactate dehydrogenase A (LDHA), to promote glycolysis in the cytosol. Thus, through phosphorylation of ME2fl, AKT1 enhances the glycolytic capacity of tumor cells in vitro and in vivo, revealing an unexpected role for subcellular translocation switching of ME2 mediated by AKT1 in the metabolic adaptation of tumor cells to growth stimuli.

Metabolic alteration facilitates autonomous proliferation and survival of tumor cells[1,2]. One of the most characteristic features of tumor cells is that they prefer to use glycolysis to produce energy and building blocks even in the presence of sufficient oxygen, which is also known as the Warburg effect[2,3]. Clearly, understanding how this metabolic change is regulated and achieved in cancer cells could reveal metabolic vulnerabilities. Although an increasing number of studies suggest that active glycolysis in tumor cells is closely linked to genetic alterations, the underlying regulatory mechanisms remain unclear.

Malic enzyme (ME) is the enzyme that catalysis the oxidative decarboxylation of malate to produce pyruvate and NADPH or NADH[4,5]. Three ME isoforms have been identified in mammalian cells: cytoplasmic NADP+-dependent isozyme (ME1), mitochondrial NAD/ P+-dependent isozyme (ME2), and mitochondrial NADP+-dependent isozyme (ME3), with ME1 and ME2 being the major isoforms[6]. By recycling the TCA cycle intermediate malate to pyruvate, these enzymes fine-tune the TCA flux, allowing the cell to maintain a certain balance of energy, reducing equivalents and biosynthetic precursor requirements. Notably, ME2

[1]Guangdong Provincial People's Hospital (Guangdong Academy of Medical Sciences), South Medical University, Guangzhou 510080, China. [2]State Key Laboratory of Molecular Oncology, School of Life Sciences, Tsinghua University, Beijing 100084, China. [3]Tsinghua-Peking Center for Life Sciences, Beijing 100084, China. [4]Shenzhen Institute of Synthetic Biology, Shenzhen Institutes of Advanced Technology, Chinese Academy of Sciences, Shenzhen 518055, China. [5]School of Life Sciences, Peking University, Beijing 100084, China. [6]State Key Laboratory of Common Mechanism Research for Major Diseases, Haihe Laboratory of Cell Ecosystem, Department of Cell Biology, Institute of Basic Medical Sciences Chinese Academy of Sciences, School of Basic Medicine Peking Union Medical College, Beijing 100005, China. [7]These authors contributed equally: Taiqi Chen, Siyi Xie. ✉e-mail: hongwu@pku.edu.cn; pengjiang@tsinghua.edu.cn; wenjingdu@ibms.pumc.edu.cn

activity is highly elevated in tumor cells and correlates with tumor progression[6–10], and NADPH generation and redox control capabilities confer potent oncogenic function to ME2[6,11]. Interestingly, loss of ME2 in pancreatic ductal adenocarcinoma (PDAC) has been shown to require ME3 for metabolic compensation and survival[12]. Recent studies have also revealed a role for ME2 in metabolically controlling mutant p53 stability and epigenetic programming[10,13].

As one of the most common events in human tumors, aberrant activation of the phosphoinositide 3-kinase (PI3K)-AKT signaling network exerts multiple effects on cellular metabolism through direct or indirect regulation, resulting in a disconnection of cell proliferation and survival from exogenous growth stimuli[14]. In general, recruitment of phosphoinositide 3-kinase (PI3K) to the plasma membrane leads to phosphorylation of phosphatidylinositol 4,5-bisphosphate (PtdIns(4,5)P2) to phosphatidylinositol 3,4,5-trisphosphate (PtdIns(3,4,5)P3), which further recruits the serine/threonine protein kinase AKT to the plasma membrane for phosphorylation at T308 and S473 by phosphoinositide-dependent protein kinase 1 (PDPK1) and mammalian target of rapamycin complex 2 (mTORC2), respectively. PtdIns(3,4,5)P3 can be dephosphorylated to PtdIns(4,5)P2 by phosphatases and Tensin homologues (PTEN), thereby attenuating PI3K-AKT signaling. Interestingly, AKT is able to be activated by SETDB1-mediated lysine methylation after PtdIns(3,4,5)P3 interaction[15,16].

In response to stimulation by growth factors or carcinogens, AKT tends to be activated in tumor cells and phosphorylates distinct substrates to perform different biological processes, including metabolism[14,17]. For instance, activated AKT directly phosphorylates NAD kinase (NADK) to produce NADP[+], facilitating tumor cell proliferation[18]. Notably, activation of AKT has been shown to be sufficient to promote aerobic glycolysis[19,20]. Through phosphorylation, AKT controls both glucose uptake and several steps in glycolysis via post-translational and also transcriptional regulation of glucose transporters and glycolytic enzymes. However, many of these regulatory effects of AKT on the promoting aerobic glycolysis that contribute to the synthesis of macromolecules in tumor cells are condition-dependent. Importantly, glycolysis is directly coupled with mitochondrial metabolism such as the TCA cycle, which acts as a major process supporting the energetic and biosynthetic demands of proliferating cancer cells[21,22]. It remains still unclear how PI3K-AKT signaling coordinates these two fundamental processes to meet energy and biosynthetic demands, and notably, the direct control of the PI3K-AKT pathway on the TCA cycle has not been determined.

Here we report that activated AKT1 directly phosphorylates ME2 at serine 9 in the cytosol, leading to a discovery of a cytoplasmic form of ME2, ME2fl. In contrast to the direct role of mitochondrial ME2 in regulating the TCA cycle, ME2fl bridges multiple key glycolytic enzymes together, resulting in a strong enhancement of glycolytic flux. Thus, through phosphorylation of ME2fl, AKT induces a metabolic switch to glycolysis from mitochondrial TCA cycle, thereby supporting tumor growth in vitro and in vivo.

## Results

### Increased cytoplasmic localization of ME2 induced by PTEN loss

We previously discovered that malic enzymes (MEs), in particularly, ME2 functions in modulating wild-type and mutant p53 activity and cell fate via distinct mechanisms[6,10]. To further explore the regulatory role of ME2 in tumors, we sought to investigate if subcellular localization would determine the metabolic activity of ME2. Previous studies have generally suggested that ME2 is a mitochondrial localization protein involved in the regulation of tricarboxylic acid cycle metabolism within mitochondria. Consistent with this, electron microscopy immunogold analysis showed that ME2 is almost exclusively localized in mitochondria in multiple PTEN-normally expressing cell lines including A549, H1299, PC9, and U2OS (Fig. 1a, up panels). However, substantial

amounts of ME2 were found in the cytosol of PTEN-deficient PC3 and U87 cells (Fig. 1a, bottom panel). These unexpected findings raise a possibility that PTEN status may influence the subcellular localization of ME2. To confirm this, we knocked down the expression of PTEN in PC9 and HepG2 cells respectively. Strikingly, PTEN depletion resulted in a marked accumulation of ME2 in the cytosol (Fig. 1b). Moreover, we also employed the Structured Illumination super-resolution Microscope (SIM) analysis and found that ME2 predominantly existed in cytosol when PTEN was depleted (Fig. 1c). Direct subcellular fraction uncovered that increased cytosol ME2 was observed in PTEN-depleted HepG2 cells (Fig. 1d). PTEN is a master regulator of multiple AKT-dependent and independent pathways associated with oncogenic effects[23]. Consistent with this, pharmacological inhibition of AKT using MK2206, a selective inhibitor of AKT, obviously reduced the cytoplasmic levels of ME2, and increased the mitochondrial localization of ME2 in multiple cell lines (Supplementary Figs. 1a–d, 2a and 2b). Moreover, inhibition of AKT attenuated the cytoplasmic accumulation of ME2 in PTEN-deficient cells (Supplementary Fig. 2c). Collectively, these findings indicate that PTEN-AKT axis regulates the subcellular localization of ME2.

### AKT1 binds to and phosphorylates ME2fl, a cytoplasmic ME2

Next, we investigated how AKT signaling activation accumulates cytoplasmic ME2. Among the AKT isoforms, AKT1 is the predominantly expressed isoform in most tissues[24]. We therefore focused on AKT1 here. Since AKT1, as a protein kinase, usually binds to its substrate proteins, we hypothesized that AKT1 may directly act on ME2 in the cytosol. To test possibility, we first examined whether AKT1 affects the cellular localization of ME2. Forced expression of C-terminal Flag-tagged ME2 (denoted as ME2-3'Flag) resulted in near complete mitochondrial localization, and when cells were co-transfected with AKT1, significant cytoplasmic localization of AKT1 and ME2 was observed (Fig. 2a), suggesting that AKT1 might directly interact with ME2. Since an apparent cytoplasmic localization of ME2 was observed in the situations of PTEN loss or AKT1 activation, we further determined whether ME2 is a protein of the same amino acid composition in cytoplasm and mitochondria. By LC-MS-mediated N-terminal amino acid sequence analysis of ME2 proteins purified from cytosol and mitochondria, we found that there are actually two different structural forms of ME2 protein, with 18 more amino acids at the N-terminal end of ME2 in the cytoplasm (namely as ME2fl) compared to ME2 in the mitochondria (namely as ME2m) (Fig. 2b, and Supplementary Fig. 3a, b). This result was also verified by the western blot analysis of the expression size of the proteins tagged different ends (Fig. 2c). In case of N-terminal Flag-tagged ME2 (5'Flag-ME2), only full-length ME2 (ME2fl) was detected, and if the Flag tag was at the C-terminus (ME2-3'Flag), 2 different sizes of ME2 were detected: full-length (ME2fl) and missing N-terminal 18 amino acids (referred as ME2m) (Fig. 2c). Cellular sub-localization analysis further revealed that ME2 was observed in the cytoplasm and mitochondria of cells transfected with C-terminal-labeled ME2fl, while N-terminal-labeled ME2, overwhelmingly, was found in the cytoplasm (Fig. 2d). Thus, these findings suggest that what was previously commonly thought to be ME2 is in fact the mitochondrial-localized ME2, a truncated peptide missing the N-terminal 18 amino acids, whereas full-length ME2 (ME2fl) is mostly found in the cytoplasm and can be accumulated by AKT activation.

Consistent with the co-localization data (Fig. 2a), AKT1 was able to bind to the full-length cytoplasmic ME2 (ME2fl), but not to ME2 that is missing the N-terminal 18 amino acids (Fig. 2e). This also suggests that the binding region of AKT1 on ME2fl should be located within the N-terminal amino acids 1–18 (Fig. 2e). In keeping with this, AKT1 did not bind to ME2 in the mitochondria (Fig. 2f). We also confirmed that AKT1 could strongly bound to ME2fl at both exogenous and endogenous levels (Fig. 2g, and Supplementary Fig. 4a–c). An in vitro binding assay using bacterially purified proteins demonstrated that AKT1 could form

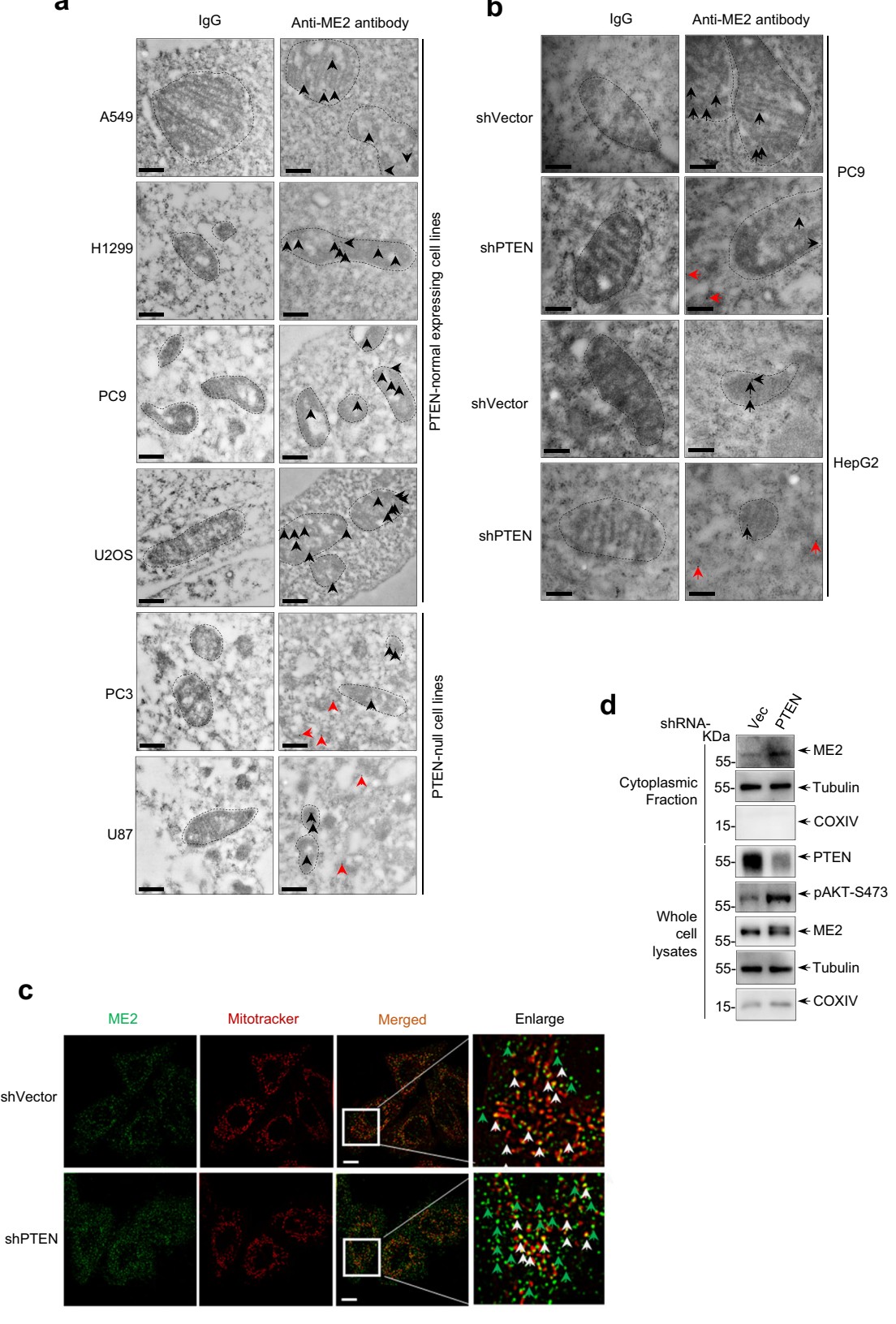

complex with ME2fl directly (Supplementary Fig. 4d). Taken together, these data suggest that AKT1 is a direct cytoplasmic binding partner of ME2fl.

AKT1 is a pivotal serine-threonine kinase determining various cellular physiological and pathological processes[14,17,25]. Cells transfected with wild-type, or a constitutively activated form (CA) of AKT1

displayed a strong phosphorylation of ME2fl (Fig. 2h, and Supplementary Fig. 4e). By contrast, introduction of the kinase-dead form (KD) of AKT1 had no effect on ME2fl phosphorylation (Fig. 2h). These findings suggest ME2fl is a phosphorylation target for AKT1. To confirm the occurrence of endogenous phosphorylated ME2 by AKT, we treated cells with various growth stimulants, including

**Fig. 1 | PTEN loss induces cytoplasmic accumulation of ME2. a, b** Electron microscopic immunogold staining of ME2 in PTEN-normal expressing (A549, H1299, PC9, U2OS) and *Pten*-null (PC3 and U87) cells (**a**), as well as in HepG2 and PC9 cells expressing vector control shRNA or shRNA targeting PTEN (**b**) using an anti-ME2 antibody (CST, cat#35939). Arrows indicate ME2 staining. Black arrows represent mitochondrial localization of ME2, and red arrows represent cytosolic or non-mitochondrial localization of ME2. Dashed circles indicate mitochondria. Scale bars, 200 nm. **c** Structured Illumination super-resolution Microscope (SIM) imaging of ME2 in HepG2 cells stable expressing control shRNA or PTEN shRNA. White arrows represent mitochondrial localization of ME2, and green arrows represent cytosolic or non-mitochondrial localization of ME2. Scale bars, 10 μm. **d** Immunoblot analysis of ME2 expression in cytoplasmic fractions and whole-cell lysates of HepG2 cells stably expressing control shRNA or PTEN shRNA. β-tubulin and COXIV served as loading controls as well as cytosolic and mitochondrial markers, respectively. All data are representative of three independent experiments.

insulin, FBS, and EGF, to activate AKT. Notably, growth stimulation led to increased phosphorylation levels of ME2fl (Supplementary Fig. 4f). Moreover, the increase in ME2fl phosphorylation induced by insulin was largely abolished by the supplementation of MK2206 (Fig. 2i). In addition to MK2206, other PI3K inhibitors, such as GDC0068, GDC0032 and PKI587, were also able to block the ME2fl phosphorylation during growth stimulation (Supplementary Fig. 4g–i). We further examined whether AKT1 directly phosphorylates ME2fl by performing in vitro phosphorylation assays using purified proteins. Remarkably, the addition of AKT1 in the presence of ATP increased the phosphorylation level of ME2fl and, interestingly, higher ME2fl enzymatic activity was observed in this case (Fig. 2j, and Supplementary Fig. 4j). These data suggest that AKT1 directly phosphorylates ME2fl and increases its enzymatic activity. In line with this, significantly increased ME2 activity was found in cells treated with insulin, and this effect was blocked by AKT inhibition (Fig. 2i).

### Serine 9 on ME2fl is responsible for the phosphorylation by AKT1

To further understand the phosphorylation of ME2fl by AKT1, we performed mass spectrometry analysis to identify the exact residue(s) phosphorylated by AKT1 on ME2fl. Specifically, we found that the peptides containing phosphorylated serine 9 (Ser 9) at the N-terminal of ME2fl were enriched (Fig. 3a). Moreover, this Ser 9 is in a conserved recognition sequence for AKT phosphorylation and, similar sequences exist in mice and rats (Fig. 3b). Thus, the residue Ser 9 might be a potential target for phosphorylation by AKT1. To confirm this, we generated a mutant ME2fl containing a serine-to-alanine substitution (S9A). Strikingly, S9A mutation completely blocked the phosphorylation of ME2fl by exogenous AKT1 (Fig. 3c). Moreover, stimulation of cells with growth factors insulin, EGF or FBS, failed to induce phosphorylation of ME2flS9A (Fig. 3d, and Supplementary Fig. 5a, b). These findings were also obtained in vitro using purified proteins. While AKT1 addition triggered the phosphorylation of wild-type ME2fl, no clear phosphorylation of ME2flS9A was observed even in the presence of AKT1(Fig. 3e, and Supplementary Fig. 5c–e). Therefore, it appears that AKT1 phosphorylates ME2fl at Ser 9. To further confirm these findings, we generated an anti-ME2fl (phospho-S9) antibody (Supplementary Fig. 5f), and again, we found that while AKT1 strongly promoted ME2fl phosphorylation when they were co-expressed in 293T cells, S9A mutation abolished the phosphorylation of ME2fl by AKT1 as detected by using the anti-ME2fl (phospho-S9) antibody, as well as an anti-RxxpS/T antibody (Fig. 3f). Similar findings were observed in in vitro phosphorylation assays using purified proteins (Fig. 2j). Importantly, phosphorylation of endogenous ME2fl was observed in response to growth stimulation, and AKT inhibition abrogated this effect (Fig. 3g). In keeping with the findings that phosphorylation increases ME2fl activity (Fig. 2i, j), the S9A mutation resulted in a decrease in ME2fl activity, whereas the serine-to-aspartate (S9D) mutation, which mimics phosphorylation, enhanced the enzymatic activity of ME2fl (Fig. 3h). Interestingly, S9A mutation enhanced the binding between ME2fl and AKT1, and ME2flS9D almost showed no binding ability towards AKT1, indicating AKT1 may prefer unphosphorylated ME2fl for interaction (Supplementary Fig. 5g). Collectively, these findings suggest that AKT1 phosphorylates ME2fl at Ser 9 and enhances its activity.

### Ser 9 phosphorylation attenuates the mitochondrial translocation of ME2fl

Increased expression or activation of AKT1 leads to cytoplasmic accumulation of ME2fl. We therefore wanted to know if phosphorylation of ME2fl forces its cytoplasmic localization. By analyzing the N-terminal of ME2fl using the PrediSi prediction method, we found that the sequence of residues 1–18 had a mitochondrial localization signal sequence feature and might be a mitochondrial localization signal peptide (Supplementary Fig. 6a). In supporting of this, deletion of this 18aa peptide leads to cytoplasmic anchoring of ME2fl (Fig. 4a, b). Moreover, disruption of the mitochondrial localization signal sequence signature by point mutations at the Arg 4 and Arg 6 sites apparently prevented GFP mitochondrial localization (Supplementary Fig. 6b, c).

In addition, if this 18-amino acid-peptide segment is typically a mitochondrial localization signal peptide, it will be processed and excised upon entry into the mitochondria. To test this, we expressed wild-type as well as Ser 9 mutant ME2fl in cells with Flag tags at their C-termini. Notably, by immunofluorescence and western blot analysis, wild-type ME2fl and S9A mutant were observed almost exclusively in mitochondria (Fig. 4c, d, and Supplementary Fig. 6d, e). In contrast, ME2flS9D exhibited both cytoplasmic and mitochondrial localizations (Fig. 4c, d, and Supplementary Fig. 6d, e). We obtained similar findings when expressing the C-terminal GFP-tagged ME2fl (Supplementary Fig. 6f, g). Thus, these findings suggest that the vast majority of ME2fl expressed in the cytoplasm may be translocated to the mitochondria and subsequently its N-terminal 1-18aa signal peptide is removed, while Ser 9 phosphorylation apparently prevents this mitochondrial localization process. To further confirm this, we constructed C-terminal GFP fusion expression plasmids expressing only the wild-type or mutant 1-18aa peptide (P1-18), and examined the subcellular localization of these fusion proteins by fluorescence microscopy. Like wild-type peptide-fused GFP proteins, GFP proteins with S9A mutant peptide exhibited almost complete mitochondrial localization (Fig. 4e, f, and Supplementary Fig. 6h, i). Strikingly, inhibition of mitochondrial protein translocation by treatment with carbonyl cyanide-m-chlorophenylhydrazine (CCCP) or dequalinium chloride (DECA), whether wild-type or mutant ME2fl, or GFP fused with the S9A mutant peptide, resulted in blocked mitochondrial localization of these proteins (Supplementary Fig. 7a–d). Consistent with the above findings, co-expression with AKT1 led to cytoplasmic accumulation of wild-type ME2fl, but not the S9A mutant (Fig. 4g). Furthermore, the interaction between AKT1 and ME2fl was able to be detected in the cytoplasm but not in the mitochondrial fraction (Fig. 2f). Taken together, these findings suggest that Ser 9 phosphorylation of ME2fl by AKT1 prevents ME2fl mitochondrial translocation, anchoring ME2fl in the cytoplasm.

### Phosphorylation of ME2fl induces a metabolic switch towards glycolysis

PTEN silencing in both HepG2 cells and PC9 cells resulted in increased glycolysis and decreased TCA cycle activity, as revealed by isotope tracing experiments (Supplementary Fig. 8a–f). This, together with the unexpected discovery of cytoplasmic ME2fl, prompted us to investigate whether it has important metabolic regulatory functions. To this end, we conducted immunoprecipitation/mass spectrometry analysis

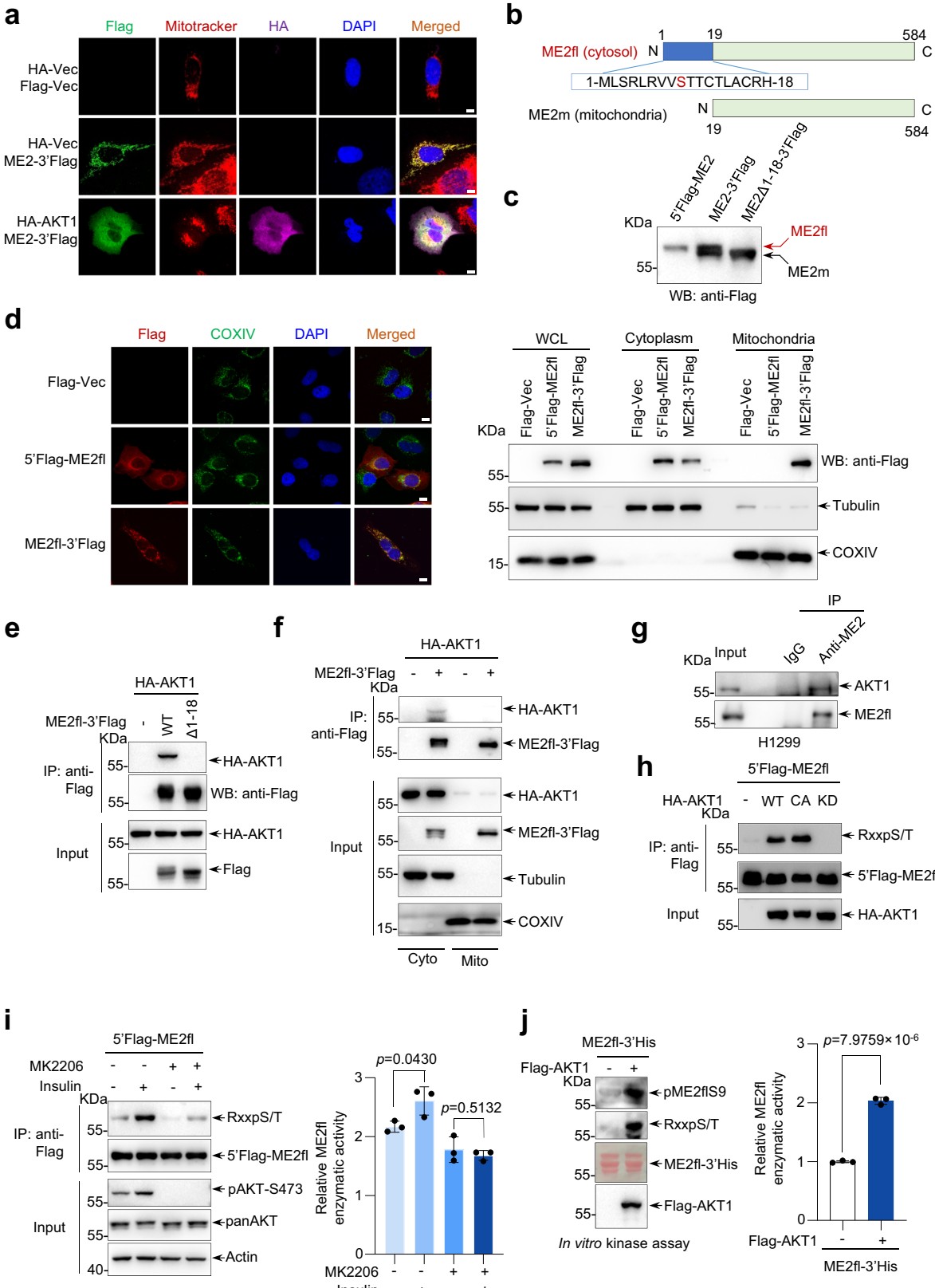

and found that overexpressed ME2fl could form complexes with glycolytic enzymes ENO1, PKM, GAPDH, and ALDO, as well as LDHA (Table 1 and Supplementary Data 1). Furthermore, analysis of the immunoprecipitants by western blotting revealed that ME2flS9D was able to bind to more than half of the glycolytic enzymes, except for PGAM, TPI, and PGK1 (Fig. 5a).

To examine whether this occurs endogenously, we developed an anti-ME2fl antibody that specifically recognizes ME2fl but not ME2m, as verified by immunoprecipitation, gene silencing and mitochondrial translocation inhibition experiments (Supplementary Fig. 9a–e). Notably, interactions between ME2fl and glycolytic enzymes PFKL, GAPDH, PKM2 and LDHA were observed in cytosol at endogenous level

**Fig. 2 | Phosphorylation of the N terminus of ME2fl by AKT1 increases cytoplasmic accumulation of ME2 and enhances ME2 activity. a** U2OS cells transfected with indicated plasmids were immunostained with an anti-Flag or anti-HA antibody. Mitochondria and DNA were stained with MitoTracker and DAPI respectively. Scale bars, 10 μm. **b** Schematic representation of cytoplasmic ME2 (ME2fl) and mitochondrial ME2 (ME2m). A mitochondrial targeting sequence (MTS) is shown at the N-terminal of ME2fl. **c** HEK293T cells transfected with indicated plasmids for 24 h were probed with an anti-Flag antibody. **d** U2OS and HEK293T cells transfected with indicated plasmids were analyzed by confocal imaging and western blotting, respectively. Scale bars, 10 μm. **e** Total lysates or anti-Flag immunoprecipitants from HEK293T cells transfected with indicated plasmids for 24 h were analyzed by Western blot. **f** Cytoplasmic and mitochondrial fractions of HEK293T cells transfected with indicated plasmids were immunoprecipitated with an anti-Flag antibody followed by western blot analysis. β-tubulin and COX IV were used to assess purity of cytosolic and mitochondrial fractions respectively. **g** Lysates of H1299 cells were immunoprecipitated with an anti-ME2fl antibody or isotype-matching control antibody (IgG) followed by western blot analysis. **h** Total lysates or anti-Flag immunoprecipitants from HEK293T cells transfected with indicated plasmids for 24 h were probed with an antibody that recognizes a minimal AKT substrate consensus RXXpS/T (hereafter referred to as anti-RXXpS/T antibody). WT wild type, CA constitutional active, KD kinase dead. **i** HEK293T cells transfected with 5′Flag-ME2fl were serum-starved for 24 h, then treated with DMSO (-) or 5 μM MK2206 for 4 h, followed by insulin stimulation for another 30 min. Cells were immunoprecipitated with an anti-Flag antibody and analyzed by Western blot. ME2fl activity was measured (shown are the means ± SD of $n = 3$ biologically independent experimental repeats for each group). **j** Purified recombinant His-tagged ME2fl (ME2fl-3′His) proteins incubated with Flag-AKT1 proteins in vitro in the presence of ATP for 30 min was probed with the anti-RXXpS/T antibody and anti-ME2fl (phospho-S9) antibody (anti-pME2flS9 antibody) (left). The activity His-ME2fl was measured (right, shown are the means ± SD of $n = 3$ biologically independent experimental repeats for each group). Data in (**a**), (**d–h**) are representative of three biologically independent experiments. Data in (**i**) and (**j**) are means ± SD, two-tailed Student's $t$ test.

(Fig. 5b). In agreement with this, forced expression ME2flS9D resulted in augmented enzymatic activities of these enzymes in 293T cells (Supplementary Fig. 9f). Similarly, HCT116 cells stably expressing ME2flS9D displayed increased enzymatic activities of PFKL, GAPDH, PKM2 and LDHA (Fig. 5c). Thus, these findings indicate that the cytoplasmic ME2fl may have a role in modulating glycolysis via acting on its multiple enzymes. Indeed, compared to those transfected with wild-type ME2fl or S9A mutants, cells forcibly expressing ME2flS9D exhibited strongly increased glycolytic flux as determined by measuring extracellular acidification rate (ECAR) under conditions where glucose was sequentially supplied to the cells to promote glycolysis, the ATP synthase inhibitor oligomycin to drive glycolysis to its maximum capacity, and the glucose analogue 2-deoxyglucose (2-DG) to block glycolysis (Fig. 5d). In addition, when cells expressed ME2flS9D, a reduction in their mitochondrial respiratory function was observed as measured by oxygen consumption rate (OCR) (Fig. 5e). Similar findings were observed in 293T cells expressing wild-type or mutant ME2fl (Supplementary Fig. 9g). In keeping with these findings, enforced expression of ME2flS9D, not the wild-type or the S9A mutant ME2fl, resulted in an elevation in the production of lactate and pyruvate (Supplementary Fig. 9h). These findings were further confirmed by the use of $^{13}$C-metabolic flux analysis. Compared to vector control cells or cells expressing wild-type ME2fl or ME2flS9A, ME2flS9D-expressing cells showed an increased yield of $^{13}$C-labeled glycolytic intermediates derived from [U-$^{13}$C$_6$]glucose and a decreased detection of $^{13}$C-labeled TCA cycle metabolites when cells were cultured with [U-$^{13}$C$_5$]glutamine (Supplementary Fig. 10a, b). Interestingly, the enhancement of glycolysis by ME2fl phosphorylation appears to be independent of ME1, as ME1 depletion showed little effect (Supplementary Fig. 11a, b).

Interestingly, in addition to increasing NADH production by promoting the glycolytic pathway, the S9D mutation also enhanced the dehydrogenase capacity of ME2, resulting in more NADPH production (Fig. 5f, g). Generally, NADPH can be used for reductive biosynthetic reactions and neutralization of reactive oxygen species (ROS). In keeping with this, a substantial reduction in ROS levels was observed in cells expressing ME2flS9D compared to those expressing wild-type ME2fl (Fig. 5h). In addition, low PH caused by increased lactate production may also contribute to ROS accumulation. Taken together, these results indicate that phosphorylation of ME2fl triggers a switch of cellular metabolism toward glycolysis and promotes the Warburg effect.

## ME2fl assembles a glycolytic enzyme complex that enhances catalytic efficiency

To gain further insight into how ME2fl promotes glycolytic activity, we investigated the possibility of ME2fl assembly of complexes involving these glycolytic enzymes, leading to an increase in the catalytic efficiency of these enzymes. Immunoprecipitation analysis revealed that ME2fl could form complexes with multiple glycolytic enzymes including PFKL, GAPDH and PKM2, as well as a glycolytic-related enzyme LDHA in HEK293T cells (Fig. 5a), and in vitro using recombinant proteins (Supplementary Fig. 12a). When ME2fl was overexpressed in HEK293T cells, the S9A mutation prevented ME2fl from binding to any of these enzymes (Supplementary Fig. 12b).

Interestingly, co-precipitation of ME2fl with each of these identified glycolytic enzymes was found by sequential co-precipitation at both exogenous (Supplementary Fig. 12c–g) and endogenous levels (Fig. 5b, and Supplementary Fig. 12h–j). Notably, silencing of ME2 strongly reduced the co-precipitation between these enzymes (Fig. 6a–c). Subcellular distribution analyses of the glycolytic enzymes using confocal microscopy further showed that there was significant colocalization between PFKL, GAPDH, PKM2, and LDHA, respectively, and this effect was remarkably abolished in cells depleted of ME2 (Fig. 6d). Moreover, analyses using density gradient centrifugation revealed that PFKL, GAPDH, PKM2, and LDHA were recovered within the same high-mobility fractions, but these enzymes dispersed when cells were depleted of ME2 (Fig. 6e, fractions 8 and 9). These findings suggest that there is a high-molecular-weight complex between these glycolytic enzymes, whose stoichiometry may be largely determined by ME2fl. In keeping with these findings, in prostate cancer tissues derived from *Pb-Cre$^+$Pten$^{L/L}$* mice, which were PTEN-deficient and displayed accumulated ME2fl, co-precipitation of ME2fl with these glycolytic enzymes was also observed (Fig. 6f).

The above findings suggest that ME2fl has an ability to facilitate the association between PFKL, GAPDH, PKM2, and LDHA. Consistent with this, in PTEN-depleted cells, in which ME2fl was accumulated, an increase in the catalytic activity of these glycolytic enzymes was observed (Supplementary Fig. 12k). We further wanted to know if ME2 could promote glycolytic reactions in vitro by incubating the recombinant enzyme mixture with glucose-6-phosphate, adenosine diphosphate (ADP), and nicotinamide adenine dinucleotide (NAD). ME2fl addition resulted in increased ATP, and intriguingly, decreased lactate production (Supplementary Fig. 12l). This unexpected finding could be due the non-canonical catalytic function of ME2 which produces 2-hydroxyglutarate (2-HG) by consuming the low pyruvate generated[10]. Indeed, when ME2fl was added to the reaction mixture, a decrease in pyruvate levels and an increase in 2-HG production were observed (Supplementary Fig. 12m). Nevertheless, these data indicate that ME2fl facilitates formation of a multi-protein complex of glycolytic enzymes, resulting in enhanced catalytic efficiency.

## Phosphorylation of ME2fl by AKT1 promotes tumorigenesis
To examine the effect of cytoplasmic ME2fl on oncogenic growth, we evaluated anchorage-independent growth in soft agar medium.

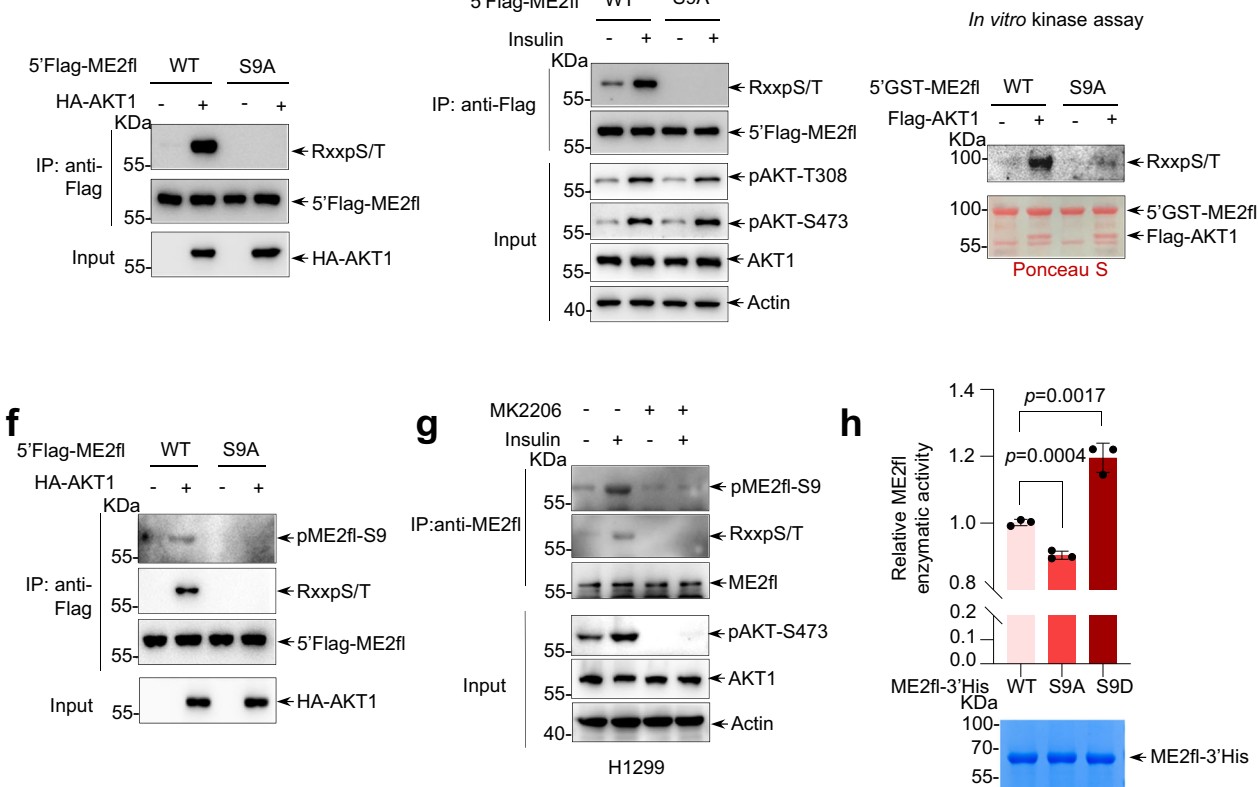

Overexpression of ME2flS9D increased anchorage-independent growth in soft agar (Supplementary Fig. 13a), an in vitro measure of tumorigenicity.

To directly investigate the role of ME2fl phosphorylation in the growth of tumor cells in animals, we injected MDA-MB-231 cells knocked-in with wild-type, S9A or S9D mutant ME2fl into immune-compromised mice. As shown in Fig. 7a, expression of ME2flS9D significantly increased the ability of the cells to produce tumors. Similar findings were obtained using HCT116 cells expressing wild-type, S9A or S9D mutant ME2fl (Supplementary Fig. 13b). Consistent with the findings that ME2fl accumulation increases glycolytic flux, ME2flS9D tumors displayed higher levels of glycolysis as probed

**Fig. 3 | AKT1 phosphorylates ME2fl at serine 9 and enhances its activity. a** Mass spectrometry analysis of immunoprecipitated 5′Flag-ME2fl from 293T cells expressing 5′Flag-ME2fl and HA-AKT1 or vector control showed an α-disintegrin fragment at m/z 199.144 (+2) matched to the charged peptide VV(pS)TTCTLACR. This phosphorylation was detected only in ME2fl expressed together with AKT1 and was confirmed by three independent experiments. **b** Sequence alignment of AKT substrate motif (R-X-R-X-X-S/T) from multiple species. **c** Lysates from HEK293T cells transfected with 5′Flag-ME2fl (WT) or ME2fl dephosphomimic mutant ME2flS9A (5′Flag-ME2flS9A) together with HA-AKT1 or vector control were immunoprecipitated with an anti-Flag antibody and phosphorylation of ME2fl was analyzed by immunoblotting. **d** HEK293T cells transfected with 5′Flag-ME2fl (WT) or 5′Flag-ME2flS9A were serum-starved for 24 h and then treated without (-) or with insulin for 30 min. Whole-cell lysates (input) and anti-Flag immunoprecipitants were analyzed by western blot for ME2fl phosphorylation using the anti-RXXpS/T antibody. **e** Purified recombinant 5′GST-ME2fl or 5′GST-ME2flS9A proteins were incubated with Flag-AKT1 in vitro in the presence of ATP for 30 min, ME2fl phosphorylation was determined by immunoblotting using the anti-RXXpS/T antibody. Purified proteins were analyzed by SDS-PAGE, followed by Ponceau S staining. **f** Total lysates or anti-Flag immunoprecipitants from transfected HEK293T cells expressing 5′Flag-ME2fl (WT) or 5′Flag-ME2flS9A together with HA-AKT1 or vector control were analyzed by immunoblotting for ME2fl phosphorylation at S9 using the anti-RXXpS/T antibody and also the anti-pME2flS9 antibody that specifically recognizes phosphorylation of serine 9 as indicated. **g** H1299 cells were serum-starved for 24 h and then treated with or without 5 μM MK2206 for 4 h before being restimulated with insulin or left unstimulated for 30 min. Whole-cell lysates and anti-ME2fl immunoprecipitants were analyzed by immunoblotting for ME2fl phosphorylation with the anti-RXXpS/T and the anti-pME2flS9 antibodies respectively. **h** The enzymatic activity of bacterially purified recombinant ME2fl-3′His (WT), ME2flS9A-3′His and ME2fl phosphomimic ME2flS9D-3′His proteins were measured respectively and expression of the indicated proteins was analyzed by SDS-PAGE, followed by Coomassie Brilliant Blue (CBB) staining. Shown are the means ± SD of $n = 3$ biologically independent experimental repeats for each group, two-tailed Student's $t$ test. All immunoblotting data are representative of three independent experiments.

with 2-DG-750 (2-deoxyglucose-750) uptake (Fig. 7b). The 2-DG fluorescence intensity measures glucose uptake and phosphorylation. Consistent with this, liquid chromatography-mass spectrometry metabolomics analysis showed a significantly increased levels of pyruvate and lactate in tumors derived ME2flS9D-expressing MDA-MB-231 cells (Fig. 7c).

To further explore the role of ME2fl phosphorylation by AKT1 in tumorigenesis in vivo, we used age-matched and genetic background-matched $Pb$-$Cre^-Pten^{L/L}$ mice as controls to monitor tumor development in the $Pb$-$Cre^+Pten^{L/L}$ prostate cancer model[26,27] (Supplementary Fig. 13c). Compared to control animals, $Pb$-$Cre^+Pten^{L/L}$ mice developed a severe prostate cancer phenotype 9 weeks after birth (Fig. 7d, e), and IHC staining showed that the levels of S473 phosphorylation of AKT1 were increased in cancer tissues of $Pb$-$Cre^+Pten^{L/L}$ mice (Fig. 7f). Notably, expression of ME2fl and phosphorylated ME2fl was clearly higher in $Pb$-$Cre^+Pten^{L/L}$ cancer tissues related to those of control mice (Fig. 7f). Moreover, $Pb$-$Cre^+Pten^{L/L}$ tumors showed higher levels of glycolysis (Fig. 7g) and increased enzymatic activity of the ME2fl-structurally associated glycolytic enzymes PFKL, GAPDH, and PKM2, as well as LDHA (Fig. 7h). Collectively, these findings demonstrate that phosphorylation and cytoplasmic accumulation of ME2fl is strongly tumorigenic and is likely an important mechanism underlying the effect of AKT1 in tumor cells.

In addition, knocking down PTEN led to strong elevation in tumor growth (Supplementary Fig. 13d, e). Yet, silencing of ME2 apparently reduced the tumor growth derived form PTEN-depleted cells (Supplementary Fig. 13d, e), further reinforcing the importance of ME2 in AKT1-mediated tumor growth.

## Discussion

ME2 converts malate to pyruvate and is required for NADPH production. ME2 is an oncoprotein that promotes tumor cell proliferation and limits cellular senescence in the context of the genetic background of wild-type p53[6]. In tumor cells bearing p53 mutations, ME2 maintained the stability of the mutant p53 via 2-HG[10]. Interestingly, in pancreatic ductal adenocarcinoma (PDA), ME2 is lost concurrently with SMAD4 deletion in ~25% of cases[12,28]. Here, we discovered a previously unrecognized mechanism by which tumor cells coordinate glycolysis and the mitochondrial TCA cycle via AKT1-mediated ME2 phosphorylation. It has been long recognized that ME2 is a $NAD^+/NADP^+$-associated mitochondrial metabolic enzyme that functionally adjusts the TCA cycle by converting malate, a TCA cycle intermediate, to pyruvate. Activation of AKT1 by growth signaling stimulation led us to the discovery of functional ME2 in the cytoplasm (namely as ME2fl). AKT1 phosphorylates ME2fl at serine 9, preventing ME2fl from translocating to the mitochondria and thus allowing it to remain in the cytoplasm. Thus, the previously widely recognized ME2 protein is actually a

truncated form of ME2fl protein with the N-terminal signal peptide missing.

The phosphorylation site (Ser 9) on ME2 is arginine-rich and matches the known substrate targeting motif of AKT, as well as the motif of S6K (a major downstream kinase of AKT). Consistent with this, supplying cells with S6K1 inhibitors, PF4708671 or LY2584702, reduced the phosphorylation of ME2fl insulin-treated cells (Supplementary Fig. 14a). Moreover, inhibition of mTORC1 strongly suppressed S6K1 and AKT, resulting in blocked ME2fl phosphorylation (Supplementary Fig. 14a). Interestingly, when co-expressed in HEK293 cells, S6K1 formed a complex with ME2fl (Supplementary Fig. 14b, c). And, like AKT1, introduction S6K1 led to a strong phosphorylation of ME2fl (Supplementary Fig. 14d). Notably, S9A mutation almost completely abolished the phosphorylation of ME2fl by S6K1 (Supplementary Fig. 14e), indicating that Ser 9 on ME2fl is also a phosphorylation site of S6K1. Thus, S6K1 may act in concert with AKT on ME2fl, but the detailed regulatory mechanisms, as well as how ME2fl is post-translationally regulated beyond phosphorylation, need to be further investigated.

The two cysteines on ME2fl adjacent to Ser 9 (Cys12 and Cys16) appear to be oxidisable, which may affect ME2 entry into mitochondria by introducing negative charges similar to those of phosphorylation. In support of this, ME2flC12SC16S (which abolishes oxidation of both cysteines) was found almost exclusively in mitochondria, whereas the C12DC16D mutation (which mimics the oxidation of both sites followed by the introduction of a negative charge) almost completely prevented ME2fl from entering the mitochondria (Supplementary Fig. 14f, g). Intriguingly, similar to ME2fl, expression of either mutant reduced intracellular ROS levels (Supplementary Fig. 14h). Thus, oxidation of these cysteines might affect the subcellular localization of ME2fl, and possibly also redox homeostasis by balancing oxidation-mediated counteraction of ROS and suppression of TCA cycle-mediated ROS scavenging, in addition to glycolysis and NADPH synthesis. Certainly, more in-depth studies are required to verify this.

Malic enzyme pathway is a major source of NADPH in certain types of tumor cells and proliferating cells[6,29], and is responsible for more than half of the NADPH production in differentiated 3T3-L1 mouse adipocytes[13,30]. Here, we found that ME2fl phosphorylation also shows increased ability to generate NADPH, and correspondingly decreased cellular ROS levels, suggesting that ME2fl may potentate NADPH production in response to oncogenic PI3K signaling. As one of the most compelling tumor and metabolic regulators[14,31,32], AKT has been reported to play an important role in NADPH metabolism, in addition to regulating many different metabolic pathways and metabolic adaptations[14,18]. AKT1 phosphorylates NADK and promotes the production of $NADP^+$, which in turn promotes NADPH production[15]. Consistent with this study, our findings support the induction of

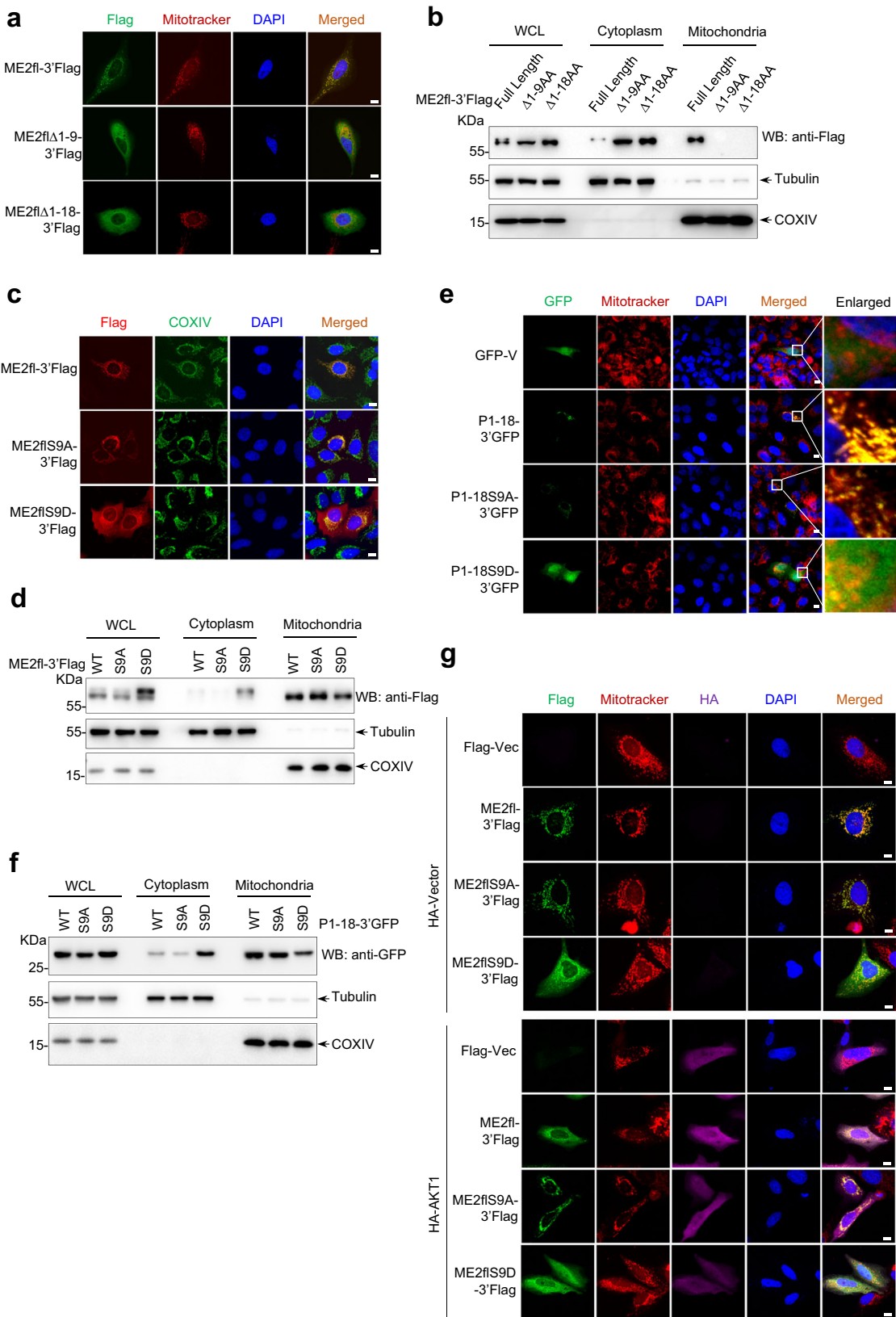

NADPH production by AKT, but through an alternative mechanism of ME2fl phosphorylation. Together, therefore, these findings suggest how AKT1 coordinates biosynthesis and antioxidant defense.

Importantly, ME2fl can form complex with multiple key glycolytic enzymes, including the rate-limiting enzymes PFKL and PKM2. By promoting the assembly of a glycolytic complex of PFKL, GAPDH, PKM2, and LDHA, ME2fl enhances the activity of these enzymes, leading to increased glycolytic flux, although the exact mechanism of action of ME2fl remains to be elucidated. These findings suggest that ME2fl may have a scaffolding role in bridging glycolytic enzymes and contribute to the Warburg effect in response to growth factor stimuli. In addition, through experiments with phosphorylation-mimicking

**Fig. 4 | Ser 9 phosphorylation by AKT1 attenuates mitochondrial translocation of ME2fl. a** Confocal imaging of U2OS cells expressing ME2fl-3′Flag or ME2fl truncates (ME2flΔ1-9-3′Flag and ME2flΔ1-18-3′Flag). ME2fl was immune-stained with an anti-Flag antibody, mitochondria were stained with MitoTracker and DNA was stained with DAPI. Scale bars,10 μm. **b** Fractionation was performed on HEK293T cells expressing ME2fl-3′Flag or ME2fl truncates (ME2flΔ1-9-3′Flag and ME2flΔ1-18-3′Flag). The expression of ME2fl and its truncations in whole-cell lysates, cytoplasmic and mitochondrial fractions were analyzed by western blotting using an anti-Flag antibody. β-tubulin and COXIV served as loading controls as well as cytosolic and mitochondrial markers, respectively. **c** Immunofluorescence analysis of ME2 in U2OS cells expressing ME2fl-3′Flag, ME2flS9A-3′Flag and ME2flS9D-3′Flag using an anti-Flag antibody. Mitochondria were stained with MitoTracker and DNA was stained with DAPI. Scale bars,10 μm. **d** HEK293T cells were transfected with ME2fl-3′Flag (WT) and ME2flS9A-3′Flag or ME2flS9D-3′Flag, followed by fractionation. The fractionations were analyzed by western blot using indicated antibodies.

β-tubulin and COXIV served as loading controls as well as cytosolic and mitochondrial markers, respectively. **e** Confocal imaging of U2OS cells expressing 3′GFP-tagged 1-18AA peptide (P1-18/WT-3′GFP) and mutant 1-18AA peptides (P1-18/S9A-3′GFP and P1-18/S9D-3′GFP). Mitochondria were stained with MitoTracker and DNA was stained with DAPI. Scale bars, 10 μm. **f** HEK293T cells expressing 3′GFP-tagged 1-18AA peptide (P1-18/WT-3′GFP) and mutant 1-18AA peptides (P1-18/S9A-3′GFP and P1-18/S9D-3′GFP) were subcellularly fractionated and analyzed by immunoblotting with an anti-Flag antibody. β-tubulin and COXIV served as loading controls as well as cytosolic and mitochondrial markers, respectively. **g** Confocal imaging of U2OS cells expressing ME2fl-3′Flag, ME2flS9A-3′Flag or ME2flS9D-3′Flag together with HA-AKT1 or HA vector control as indicated. ME2fl and AKT1 was immune-stained with the anti-Flag and anti-HA antibodies respectively, mitochondria were stained with MitoTracker and DNA was stained with DAPI. Scale bars, 10 μm. All data are representative of three independent experiments.

mutations at serine 9, we found that phosphorylated ME2fl has significantly attenuated mitochondrial metabolism, suggesting that AKT1 directly regulates the transition between intracellular glycolysis and mitochondrial metabolism by altering the phosphorylation status of ME2. These findings not only reveal a previously unappreciated cytoplasmic ME2 with an important regulatory role in glycolysis, but also demonstrate that AKT1 is a metabolic switch for tumor cell proliferation in response to growth stimuli.

In this study, we found that growth factor stimulation induces AKT-mediated phosphorylation of ME2fl, a form of ME2 that is virtually undetectable in the absence of stimulation (which is probably why we have not found this cytoplasmic form to exist to date). Phosphorylation of ME2fl on Ser 9 by AKT1 upon growth stimulation anchors ME2fl in the cytosol and the accumulation of ME2fl leads to an increase in glycolytic activity, causing a Warburg effect in cancer cells and promoting cell proliferation (Supplementary Fig. 15). In addition to its tumor-promoting effects, it may also play a role in some proliferating cells (e.g. hematopoietic stem cells, etc.), as these rapidly proliferating cells also tend to be highly glycolytic. Thus, it would be of interest to further elucidate what physiological functions this newly discovered form of cytoplasmic ME2 has in addition to its tumor-promoting effects in PTEN-deficient or AKT-activated tumors. Nevertheless, our findings provide a previously unappreciated perspective and clue to understanding the pro-oncogenic function of AKT and the Warburg effect in cancer cells.

## Methods
### Inclusion and ethics
The study is compliant with all of the relevant ethical regulations regarding animal research. All experiments involving mice are strictly complies with the protocols approved by the Association for Assessment and Accreditation of Laboratory Animal Care International and Institutional Animal Care and Use Committee (IACUC) of Tsinghua University. The laboratory animal facility has been licensed by the IACUC.

### Antibodies
Antibodies used in this study were purchased from the indicated sources: anti-Flag (Sigma, F1804; 1:10,000 dilution in immunoblotting; 1:1000), anti-HA (Bioeasytech, BE2007; 1:3000 for immunoblotting; 1:1000 for immunofluorescence), anti-GFP (MBL, 598; 1:3000), anti-panAKT (C67E7) (CST, 4691; 1:1000), anti-AKT1 (C73H10) (CST, 2938; 1:1000), anti-pAKT T308 (C31E5E) (CST, 2965; 1:1000), anti-pAKT S473 (D9E) (CST, 4060; 1:1000 for immunoblotting; 1:100 for immunohistochemistry), anti-COXIV (Proteintech, 11242-1-AP; 1:1000 for immunoblotting; 1:100 for immunofluorescence), anti-pAKT substrate RXXS/T(110B7E) (CST, 9614; 1:1000), anti-ME2 (Proteintech, 24944-1-AP; 1:1000 for immunoblotting; 1:100 for immunoprecipitation), anti-ME2 (Proteintech, 67457-1-IG; 1:100), anti-ME2 (E1N3F) (CST, 35939; 1:20 for immunogold electronic microscope; 1:400 for immunofluorescence), anti-GAPDH (Proteintech, 60004-1-Ig; 1:3000 for immunoblotting; 1:100 for immunoprecipitation), anti-PFKL (A-6) (Santa Cruz, sc-393713; 1:1000 for immunoblotting; 1:100 immunoprecipitation), anti-PKM2 (D78A4) (CST, 4053; 1:1,000 for immunoblotting; 1:100 for immunoprecipitation), anti-LDH-A (Proteintech, 19987-1-AP; 1:3000 for immunoblotting; 1:100 for immunoprecipitation), anti-GSK3β (1F7) (Santa Cruz, sc-53931; 1:1000), anti-pGSK3β-S9 (CST, 9336; 1:1000), anti-p-p70S6K-T389 (108D2) (CST, 9234; 1:1000), anti-p70S6K (CST, 9202; 1:1000), anti-pS6-235/236 (D57.2.2E) (CST, 4858; 1:1000), anti-S6 (5G10) (CST, 2217; 1:1000), anti-β-Actin (Bioeasytech, BE0037; 1:5000), anti-GST (Bioeasytech, BE2013; 1:5000), anti-Tubulin (Bioeasytech, BE0031; 1:3000), Normal mouse IgG (Santa Cruz, sc2025; 1:100), Normal Rabbit IgG (CST, 2729; 1:100), Goat anti-rabbit IgG-HRP (Bioeasytech, BE0101; 1:5000), Goat anti-mouse IgG-HRP (Bioeasytech, BE0102; 1:5000). Alexa Fluor® 488 Goat anti-Rabbit IgG (H+L) (ThermoFisher Scientific, A11008; 1:1000), Alexa Fluor® 594 Goat Anti-Mouse (ThermoFisher Scientific, 37115; 1:40), Goat anti-Mouse IgG (H + L) Cross-Adsorbed Secondary Antibody, Alexa Fluor® 647 conjugate (ThermoFisher Scientific, 32728; 1:1000), Goat anti-Rabbit IgG (H+L) Cross-Adsorbed Secondary Antibody, Alexa Fluor® 647 conjugate (ThermoFisher Scientific, 21246; 1:1000). The custom-designed polyclonal antibodies listed as followings were in cooperation with Abclonal Technology (Wuhan, China): anti-phospho-ME2fl Ser9 (p-ME2fl-S9, antigen peptide: C-SRLRVV(S-p)TT; 1:1000 for immunoblotting; 1:200 for immunohistochemistry), anti-N-ME2fl (anti-N-ME2fl, antigen peptide: SRLRVVSTTCTLACRH; 1:1000), Colloidal Gold AffiniPure Goat Anti-Rabbit IgG (H + L) (Jackson ImmunoResearch Laboratories, 111-195-144; 1:50 dilution for electron microscopic immunogold staining).

**Table 1 | Immunoprecipitation-mass spectrometry analysis of phosphorylated ME2fl-interacting proteins**

| Uniprot ID | Protein name | Score |
|---|---|---|
| P06733 | ENO1 | 46.14 |
| P14618 | PKM | 43.48 |
| P04406 | GAPDH | 34.28 |
| P04075 | ALDOA | 19.88 |
| P00338 | LDHA | 19.65 |
| A2IBT6 | G6PD | 2.38 |

Lysates from HEK293T cells expressing ME2flS9D-3′Flag or vector control were immunoprecipitated with an anti-Flag antibody, followed by mass spectrometry analysis. The bound proteins were digested and analyzed by high sensitivity LC-MS/MS using an Orbitrap Elite mass spectrometer (ThermoFisher Scientific). Fragment spectra against the UniProt protein database was searched for protein identification. Shown are the glycolytic enzymes that may interact with phosphorylated ME2fl and their scores on the mass spectrometer.

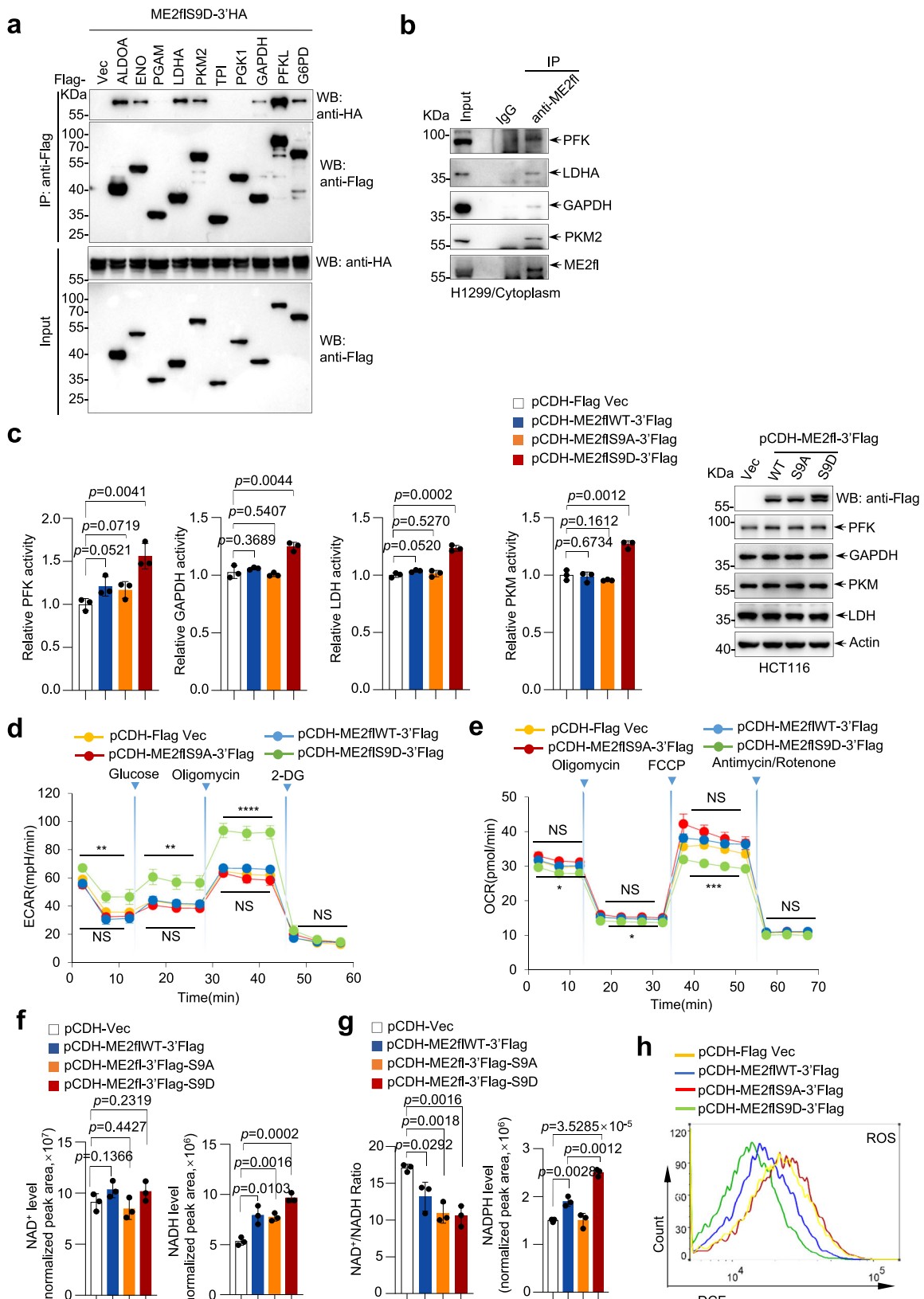

## Reagents

Reagents used in this study were purchased from the indicated sources: L-malic acid (Sigma, M1000), NAD⁺ (Sigma, N7004), MK2206 (Selleck, S1078), GDC0032 (Selleck, S7103), PKI587 (Selleck, S2628), PF4708671 (Sigma, 559273), LY2584702 (Sigma, SML2892), TritonX-100 (Sigma, X100), Imidazole (Sigma, I2399), Sucrose (Sigma,

V900116), NaCl (Sigma, S7653), PFA (Sangon, E672002), Q5® High-Fidelity DNA Polymerase (New England Biolabs, 0491L), T4 DNA ligase (New England Biolabs, M0202), APS (Sigma, A3678), TEMED (Sigma, T9281), Acryl/Bis 30% Solution (Sangon, B546017), H2-DCFDA (Sigma, 35845), Glucose (Sigma, G7528), Oligomycin (Sigma, 495455), FCCP (Sigma, C2920), 2-DG (Sigma, D8375), Antimycin A(Selleck, S1478),

**Fig. 5 | AKT1 phosphorylation of ME2fl induces a metabolic switch to glycolytic activity from mitochondrial respiration. a** Total lysates or anti-Flag immuno-precipitants from transfected HEK293T cells expressing 3'HA-tagged ME2flS9D (ME2flS9D-3'HA) together with Flag-tagged glycolytic enzymes or Flag vector control as indicated were analyzed by immunoblotting. Representative results are shown from three independent experiments. **b** Cytoplasmic fractions of H1299 cells were immunoprecipitated with an anti-ME2fl antibody and then analyzed by western blotting with the anti-PFK, anti-GAPDH, anti-PKM, and anti-LDH antibodies, respectively. Representative results are shown from three independent experiments. **c** HCT116 cells were stably infected with control lenti-virus (pCDH-Flag Vec) or lenti-viruses expressing wild-type ME2fl (pCDH-ME2fl-3'Flag), ME2flS9A (pCDH-ME2flS9A-3'Flag) or ME2flS9D (pCDH-ME2flS9D-3'Flag). The activity of the indicated glycolytic enzymes was measured (left) and protein expression was analyzed by western blotting (right). Data are means ± SD, two-tailed Student's *t* test. **d**, **e** HCT116 cells stably expressing wild-type ME2fl (pCDH-ME2fl-3'Flag), ME2flS9A (pCDH-ME2flS9A-3'Flag), ME2flS9D (pCDH-ME2flS9D-3'Flag), or vector control

(pCDH-Flag Vec) were used for ECAR (**d**) or OCR (**e**) analysis. Cells were supplied with 25 mM glucose, 1 μM oligomycin and 100 mM 2-DG at the indicated times for ECAR analysis, and 1 μM oligomycin, 1 μM FCCP, and 2.5 μM antimycin/rotenone at the indicated times for OCR analysis by using a Seahorse XFe96 analyzer system. *n* = 6 biologically independent experimental repeats for each group in (**d**) and *n* = 4 biologically independent experimental repeats for each group in (**e**). Data are means ± SD, *$P < 0.05$; **$P < 0.01$; ***$P < 0.001$; ****$P < 0.0001$; NS, no significance; two-tailed Student's *t* test. Exact *P* values are shown in Source data. **f**–**h** The abundance of $NAD^+$ and NADH, and the $NAD^+$/NADH ratio (**f**), and NADPH levels (**g**) in HCT116 cells stably expressing ME2fl (pCDH-ME2fl-3'Flag), ME2flS9A (pCDH-ME2flS9A-3'Flag), ME2flS9D (pCDH-ME2flS9D-3'Flag), or vector control (pCDH-Flag Vec) were measured by LC-MS (*n* = 3 biologically independent wells from distinct biological sources). ROS level (**h**) was determined by 2′,7′-di-chlorofluorescein (DCF) staining and flow cytometry analysis. Data are means ± SD; two-tailed Student's *t* test.

Rotenone (Sigma, R8875), Protein G sepharose beads (Abcam, ab193259), Insulin (Sigma, I0516) and EGF (Sigma, SRP3027), anti-Flag M2 affinity gel (Sigma, A2220, 1 μL for 100 μg lysates), Glutathione Sepharose 4B (Cytiva, 17-0756-01), Ni-NTA Agarose (Sangon, C600033), Carbonyl Cyanide 3-Chlorophenylhydrazone (MedChem-Express, HY-100941), Dequalinium chloride (Selleck, S4066), 3Flag-peptide (Sigma, F4799), GSH (Sangon, A100-399), 2-Deoxy Glucose (DG)-750 (PerkinElmer, 760562), Fructose-6-phosphate (Sigma, F3627), ATP (Sigma, A26209), ADP (Sigma, A2754), AMP (Sigma, 1012178), $NH_4Cl$ (Sigma, A9434), KCl (Sigma, P3911), NADH (Sigma, N8129), $Na_2HPO_4$ (Sigma, S9763), Aldolase (Sigma, A8811), α-Glycerophosphate dehydrogenase (Sigma, G6751), Triose phosphate isomerase (Sigma, T2391), Triethanolamine (Sigma, V900257), EDTA (Amersco, 0105), Glyceraldehyde-3-phosphate (Sigma, G5251), Phos-phoenol pyruvate (Sigma, 0654), Lactate dehydrogenase (Sigma, L2500), Sodium pyruvate (Sigma, P5280), [U-$^{13}C_6$]glucose(Cambridge Isotope Laboratories, CLM-1396-1), [U-$^{13}C_5$]glutamine(Sigma, 605166), DAPI (ThermoFisher Scientific, 62247), ProLong™ Gold Antifade Mountant with DAPI (ThermoFisher Scientific, P36935).

### Mice
4- to 5-week-old male athymic Balb-c nu/nu male mice were purchased from Huafukang Laboratory Animal Technology (Beijing, China) for xenograft experiment. All mice were housed in isolated ventilated cages (maxima six mice per cage) barrier facility at Tsinghua University. The mice were maintained on a 12/12-hour light/dark cycle, 22–26 °C, 50% humidity on average with sterile pellet food (Jiangsu Xietong Pharmaceutical Bioengineering Co., Ltd, XTC01FZ-003) and water ad libitum. *Pb-Cre⁺; PtenL/L* prostate cancer model was obtained by crossing *Pb-Cre⁺;Pten^{L/L}* male mice to *Pb-Cre;Pten^{L/L}* female mice. Mouse genotypes were verified by PCR with primers for Cre. The primer sequences for Cre are: Forward: 5′-GCCTGCATTACCGGTC-GATGC-3′, Reverse: 5′-CAGGGTGTTATAAGCAATCCC-3′. All animal procedures and experiments in this study are strictly in accordance with the protocols approved by the IACUC of Tsinghua University. The laboratory animal facility has been licensed by the IACUC. The study is compliant with all of the relevant ethical regulations regarding animal research.

### DNA constructions and site mutagenesis
Polymerase Chain Reaction (PCR) -amplified human wild-type ME2fl, ME2m (Δ1-18) and AKT1 were cloned into pRK5-5′Flag, 3′Flag, 5′HA or 3′HA vectors with standard cloning protocol using 2 × Q5® High-Fidelity DNA Polymerase (New England Biolabs, M0491). Coding Sequence (CDS) of ME2fl was subcloned into pRK5-3′GFP, pCDH-CMV-MCS-EF1-Puro, pET21b or pGEX-6P-1 vectors. pRK5-ME2fl-1-18aa-GFP, pET21b-ME2flS9D-3′His, pRK5-ME2fl-P1-18/2RA(R4AR6A)-3′GFP, pRK5-

ME2fl-P1-18/R17A-3′GFP, pRK5-ME2fl-P1-18/3RA-3′GFP (R4AR6A17A) were generated by primer annealing. pRK5-ME2flS9A-3′Flag, pRK5-ME2flS9A-3′HA, pRK5-ME2flS9A-3′GFP, pET21b-ME2flS9A-3′His, pGEX6P-1-5′GST-ME2flS9A, pRK5-ME2flS9D-3′Flag, pRK5-ME2flS9D-3′HA, pRK5-ME2S9D-3′GFP, pRK5-ME2flS9D-3′His and pRK5-HA-AKT1-Kinase Dead (K179M) were constructed using a site-directed muta-genesis method. The primers used are listed in Supplementary Data 2.

### Cell culture and transfections
HEK293T cells, human hepatoma HepG2 cells, human osteosarcoma U2OS cells, human colon cancer HCT116 cells, human lung carci-noma A549 cells and PC9 cells, human non-small cell lung cancer H1299 cells, human prostate adenocarcinoma PC3 cells and human glioblastoma U87MG cells were purchased from ATCC. HEK293T, HepG2, A549 and HCT116 cells were cultured in Dulbecco's modified Eagle's medium (ThermoFisher Scientific, C11995500BT) with 10% FBS (ThermoFisher Scientific, 16000044). U2OS cells were cultured in Maccoy's 5A medium with 10% FBS. PC9, H1299 and U87MG cells were cultured in RPMI-1640 medium (ThermoFisher Scientific, 11875093) with 10% FBS. All cells were cultured in a 5% $CO_2$ humi-dified incubator (ThermoFisher Scientific, Heracell 150i) at 37 °C and subjected to examination of mycoplasma contamination examined for mycoplasma contamination and cultured for no more than 2 consecutive months. None of the cell lines used in this work was listed in the ICLAC database.

Cells were seeded at 50% confluence and transfected with human ME2 or PTEN siRNAs using Lipofectamine 3000 transfection reagent (ThermoFisher Scientific, L3000015). The siRNA targeting luciferase was used as control. All siRNAs were used at a concentration of 20 nM. The siRNA sequences were listed as follows:

siLuciferase:5′-CGUACGCGGAAUACUUCGATT-3′
siME1: 5′-GGGCAUAUUGCUUCAGUUCUU-3′
siME2: 5′-GCACGGCUGAAGAAGCAUAUAUU-3′
siPTEN: 5′-GGUGUAAUGAUAUGUGCAUUU-3′

The siRNA transfection procedures were performed according to the manufacturer's instructions.

### Growth factors and inhibitors
Cells seeded at a confluence of 80% were starved in serum-free med-ium for 24 h and stimulated with insulin (100 nM), EGF (50 ng/mL) or FBS (10%, v/v) for 30 min.

For the treatment of cells with inhibitors, MK2206 (5 μM), GDC0032 (4 μM), PKI587 (1 μM) or DMSO were added into culture medium and cells were cultured for 4 h followed by grow factor sti-mulation. After washing with PBS, cells were collected by being lysed with IP Lysis buffer (50 mM Tris-HCl pH 7.4, 150 mM NaCl, 1% TritonX-100, 5 mM EDTA, 1% sodium deoxycholate) supplemented with

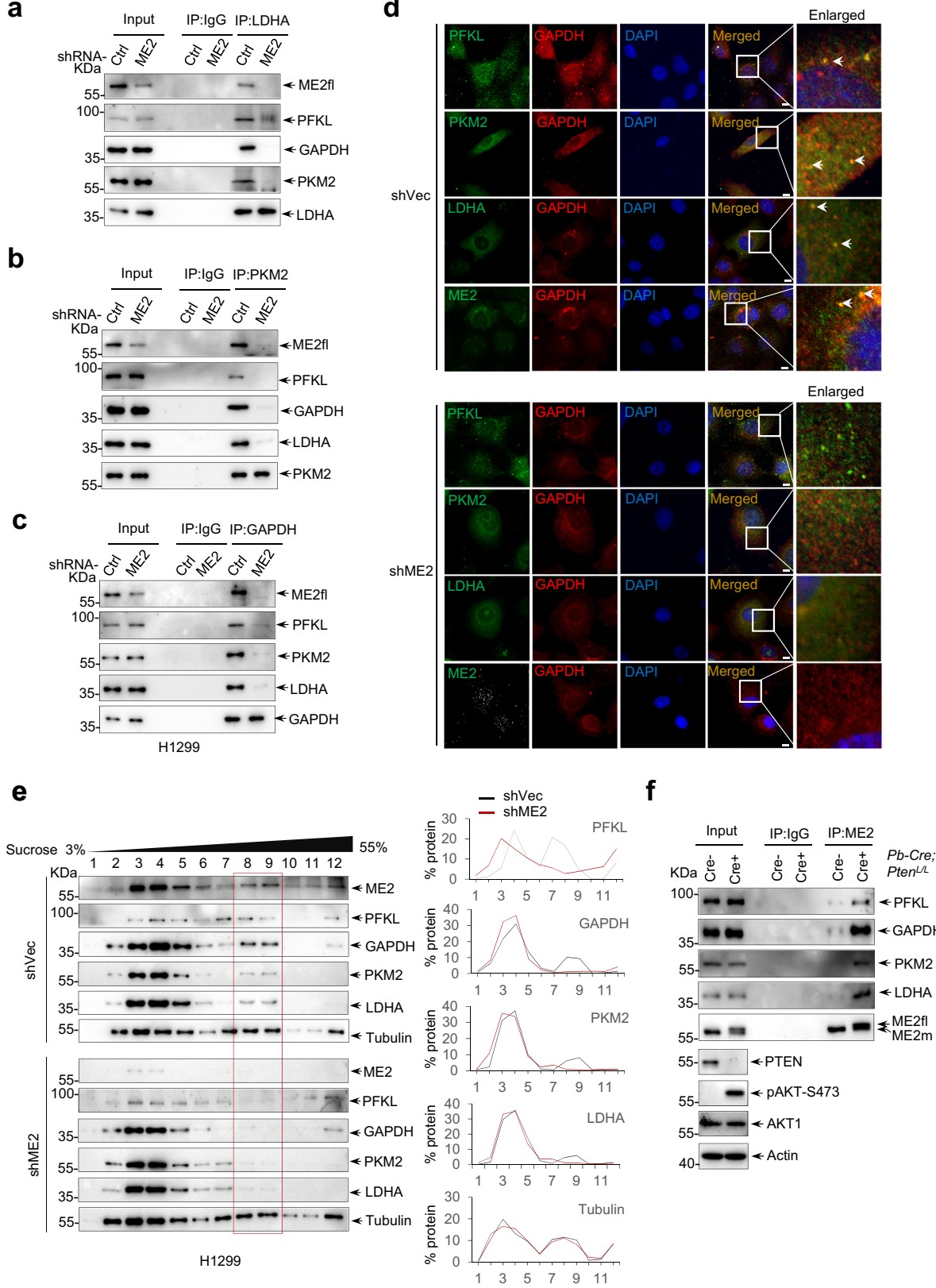

protease inhibitors (Complete cocktail, Roche) and phosphatase inhibitors (10 mM NaF, 1 mM Na₃VO₄).

For treatment of cells with mitochondrial inhibitors, cells were precultured in medium containing 10 µM CCCP or 10 µM DECA for 4 h before transfection followed by consecutively treatment till being collected.

## Cell lysis and Immunoprecipitations

Cells were lysed by using modified RIPA buffer (10 mM Tris-HCl pH 7.5, 5 mM EDTA, 150 mM NaCl, 1% NP-40, 1% sodium deoxycholate, 0.025% SDS) and protease inhibitors on ice for 30 min. Protein sample concentration was quantified using BCA protein assay kit (Sangon, 503021), boiled in 5 × SDS loading buffer, resolved by SDS-PAGE and

**Fig. 6 | ME2fl assembles a glycolytic complex in the cytosol. a–c** Cytoplasmic fractions of H1299 cells expressing vector control shRNA (shCtrl) or shRNA targeting ME2 (shME2) were immunoprecipitated with anti-LDHA (**a**), anti-PKM2 (**b**) and anti-GAPDH (**c**) antibodies. The immunoprecipitants were analyzed by western blotting using the antibodies against ME2fl, PFKL, PKM2, GAPDH, or LDHA as indicated. **d** Confocal imaging of shCtrl or shME2 H1299 cells comparing staining between GAPDH and PFKL, PKM2, LDHA or ME2. Scale bars, 10 μm. **e** Protein complexes from shCtrl or shME2 H1299 cells were separated by sucrose density gradient centrifugation and analyzed by western blotting with the indicated antibodies (left panel). The data were quantified using Image J v.1.46 (right panel). All data are representative of three independent experiments. **f** Lysis from prostate tissues of the wild type (*Pb-Cre⁻Pten^{L/L}*) and *Pten*-condition knockout (*Pb-Cre⁺Pten^{L/L}*) mice were immunoprecipitated using an anti-ME2 antibody. Whole-cell lysis (input) and the immunoprecipitations were analyzed by western blot with the indicated antibodies. All data are representative of three independent experiments.

transferred onto nitrocellulose membrane. 5% skimmed milk (BD, 232100) in TBS supplemented with 0.1% Tween (TBST) was used to block the membrane before probing with indicated antibodies in 5% BSA in TBST at 4 °C overnight. Membranes were washed with TBST and then incubated with HRP-conjugated anti-mouse or anti-rabbit secondary antibodies at room temperature for 45 min and developed with ECL Western Blotting Detection Reagent (Tanon, 180-5001).

For immunoprecipitations, cells were lysed in IP Lysis buffer (50 mM Tris-HCl pH 7.4, 150 mM NaCl, 1% TritonX-100, 5 mM EDTA, 1% sodium deoxycholate) supplemented with protease and phosphatase inhibitors for 30 min followed by centrifugation at $13,000 \times g$ for 10 min at 4 °C. The supernatants (1 mg) were incubated with the indicated antibody (1–2 μg) for 12 h at 4 °C, followed by overnight incubation with Protein A/G sepharose beads (Abcam, ab193259). Beads were washed 3 times with IP Lysis Buffer. Immunoprecipitants were eluted with 3Flag peptide for protein purification or 0.1 M glycine (pH 2.5) at $100 \times g$ for 5–10 min followed by neutralization with 100 mM Tris buffer for immunoblotting analysis.

### Purification of recombinant proteins

For Flag-tagged proteins purification, HEK293T cells were transfected with pRK-5′Flag-AKT1, pRK-5′Flag-ME2flWT or mutant ME2fls (pRK-5′Flag-ME2flS9A and pRK-5′Flag-ME2flS9D), respectively. 36 h after transfection, cells were collected and lysed in IP Lysis Buffer with protease and phosphatase inhibitors for 30 min and centrifuged at $13,000 \times g$ for 10 min at 4 °C. Flag M2 affinity gel was added into the supernatants to immunoprecipitate Flag-tagged recombinant proteins. After overnight incubation, Immunoprecipitants were washed 3 times with wash buffer (20 mM Tris-HCl, pH 7.5, 150 mM NaCl, 1% TritonX-100), followed by competitive elution with appropriate synthetic 3Flag Peptide in TBS.

For GST-tagged protein purification, pGEX-6P-1-5′GST-Vector, pGEX-6P-1-5′GST-ME2flWT, pGEX-6P-1-5′GST-ME2flS9A or pGEX-6P-1-5′GST-ME2flS9D plasmids were transformed into BL21/DE3. Transformants were amplified in 5 mL LB medium (containing 100 μg/mL ampicillin) and grown overnight at 37 °C to stationary phase. 1 mL medium with amplified transformants was then inoculated into new LB medium. The cultures were grown at 25 °C to an OD600 of ~0.6 before inducing with 0.3 mM IPTG for 12 h. Cell pellets were collected, resuspended in 50 mL lysis buffer (50 mM Tris-HCl pH 8.0, 150 mM NaCl, 1% Triton X-100, 1 mM PMSF), and lysed by sonication before centrifugation at $13,000 \times g$ for 10 min at 4 °C. Supernatants were then incubated with glutathione-agarose beads for 12 h at 4 °C. Beads were washed extensively with lysis buffer 3 times before eluting for 1 h in fresh GSH buffer (50 mM Tris-HCl pH 7.5, 30 mM GSH).

For His tagged protein purification, pET21b-ME2fl/WT-3′His, pET21b-ME2fl/S9A-3′His, pET21b-ME2fl/S9D-3′His plasmids were transformed into BL21/DE3. Transformants were induced by IPTG to produce fusion proteins as described above. Cells were lysed by sonication in Ni-NTA Lysis buffer (50 mM NaH₂PO₄, 300 mM NaCl, 10 mM Imidazole, pH 8.0). The lysates were centrifuged at $13,000 \times g$ for 10 min at 4 °C then incubated with 1 mL Ni-NTA Resin for 12 h at 4 °C. Ni-NTA Resin was washed with 10 mL wash buffer (Lysis buffer plus 20 mM imidazole) for 3 times and the recombinant proteins were eluted with 2 mL elution buffer (Lysis buffer plus 250 mM imidazole).

### GST pull-down assay

HEK293T cells were transfected with plasmids encoding Flag-tagged proteins for 24 h and cells were lysed in IP Lysis buffer (50 mM Tris-HCl pH 7.4, 150 mM NaCl, 1% TritonX-100, 5 mM EDTA, 1% sodium deoxycholate) supplemented with protease and phosphatase inhibitors. Flag-tagged proteins were immunoprecipitated with anti-Flag M2 affinity gel for 12 h and eluted by 3Flag peptide for 12 h at 4 °C. Prokaryotically expressed GST or GST fusion proteins were lysed and sonicated in lysis buffer (50 mM Tris-HCl pH 8.0, 150 mM NaCl, 1% Triton X-100, 1 mM PMSF) and incubated with glutathione agarose beads for 12 h. The glutathione agarose beads were washed for 3 times with lysis buffer and incubated with eukaryotically expressed Flag-tagged proteins in binding buffer (50 mM Tris-HCl pH 8.0, 150 mM NaCl, 1% Triton X-100, 1 mM PMSF) for 12 h at 4 °C. The glutathione agarose beads were then washed 3 times with binding buffer, boiled in 1 × SDS loading buffer and then subjected to immunoblotting with an anti-Flag antibody for Flag-tagged proteins and Coomassie Brilliant Blue staining for GST-tagged proteins.

### Immunofluorescence

Cells were plated on gelatin-coated glass coverslips (Corning) in a 12-well plate at 50–70% confluence and transfected with 1 μg indicated plasmids for 24 h. After transfection, cells were washed with PBS and fixed by 4% paraformaldehyde (PFA) for 20 min. For MitoTracker staining, cells were cultured with medium containing 100 nM Mito-Tracker at 37 °C in the incubator with 5% CO₂ for 30 min before fixation. The coverslips were washed 2 times with PBS after fixation and permeabilized in PBS containing 0.2% Triton X-100 for 20 min at room temperature. After washing with PBS for once and blocking with 5% BSA in PBS for 1 h at room temperature, cells were incubated with primary antibodies for 12 h at 4 °C, washed with PBST (PBS supplemented with 0.1% Tween) for 3 times and stained with fluorescent-dye-conjugated anti-mouse or anti-rabbit secondary antibodies (1:1000). Cells were washed 3 times after secondary antibodies incubation followed by nuclei staining (DAPI, 0.5 μg/mL) for 5 min. The coverslips were finally washed with PBS, dried at room temperature and mounted on glass slides with an anti-quench mounting buffer (ThermoFisher Scientific). Cells were imaged by FLUOVIEW FV3000 Confocal Laser Scanning Microscope (Olympus) and LSM 780 Confocal Microscope (Zeiss).

### Cell fractionation

The cell fraction assay was performed according to a previously reported method. $1 \times 10^7$ cells were homogenized in 1 mL fraction buffer (20 mM HEPES-KOH, pH 7.5, 10 mM KCl, 1.5 mM MgCl₂, 1 mM sodium EDTA buffer, 1 mM sodium EGTA buffer, 1 mM dithiothreitol and protease inhibitor cocktail) in the presence of 250 mM sucrose for 20 min on ice followed by fractionation via passing through a 27 gauge needle with 1 mL syringe for 15 times and kept on ice for another 20 min. 50 μL homogenates were transferred into new tubes as whole-cell lysis (WCL). Homogenates were centrifuged at 500 g for 5 min at 4 °C and the pellets were discarded. The supernatants were collected and centrifuged again at $10,000 \times g$ for 20 min at 4 °C to obtain cytosolic and mitochondrial fractions. Mitochondrial fractions were lysed with IP lysis buffer (50 mM Tris-HCl pH 7.4, 150 mM NaCl, 1% TritonX-

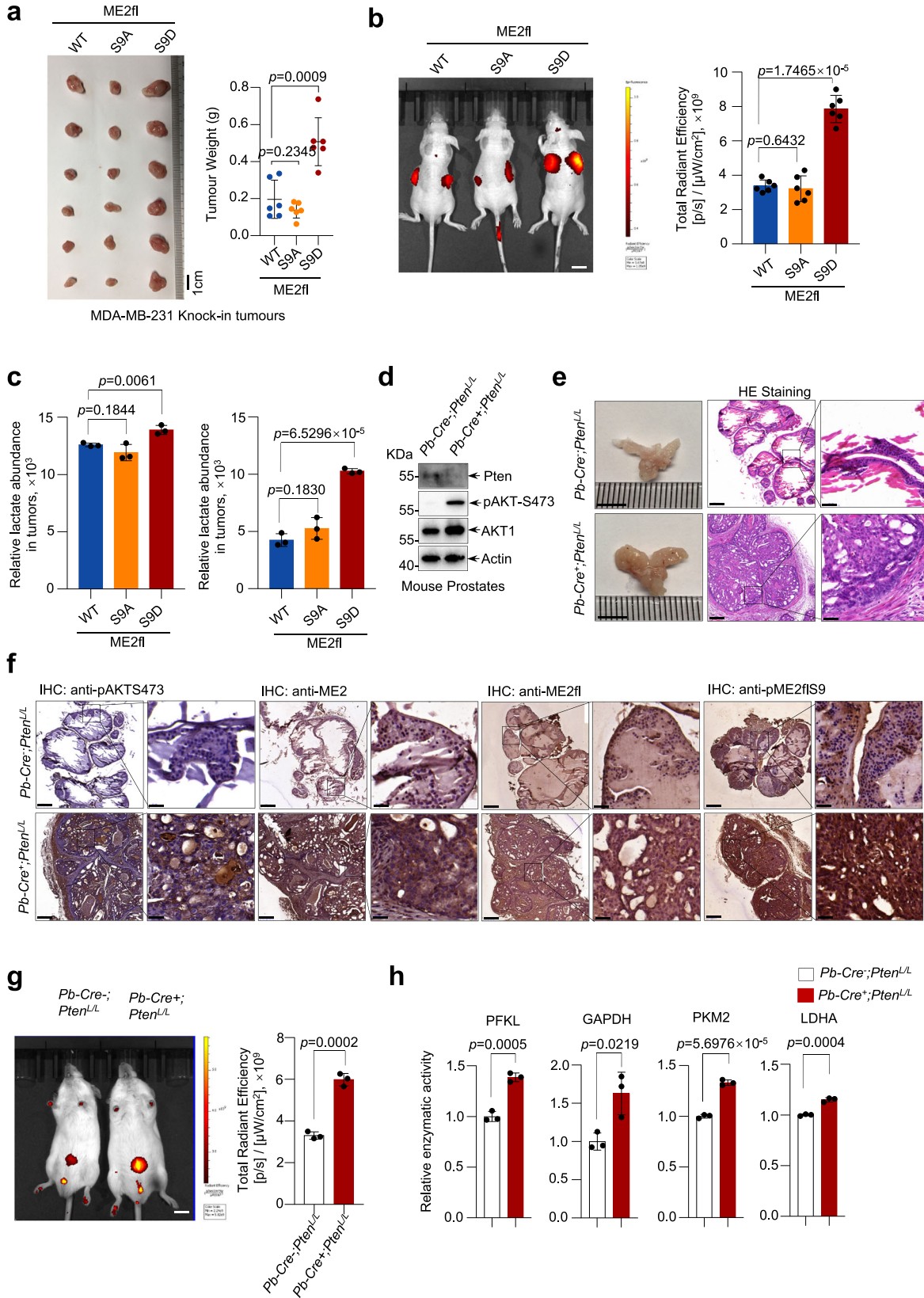

100, 5 mM EDTA, 1% sodium deoxycholate) supplemented with protease and phosphatase inhibitors on ice for 30 min. Both cytoplasmic and mitochondrial fractions were quantified using the Bradford method and boiled in 5 × SDS loading buffer. 20 µg of each fraction was analyzed by immunoblotting. β-tubulin and COXIV were markers for cytoplasm and mitochondria, respectively.

## In vitro kinase assays

HEK293T cells were transfected with Flag-tagged active AKT1 (pRK5-Flag-AKT1) for 24 h and cells were lysed in IP lysis buffer (50 mM Tris-HCl pH 7.4, 150 mM NaCl, 1% TritonX-100, 5 mM EDTA, 1% sodium deoxycholate) supplemented with protease and phosphatase inhibitors on ice for 30 min. The lysates were centrifuged at 13,000 × g for

**Fig. 7 | Ser 9 phosphorylation of ME2fl promotes tumor growth in vitro and in vivo. a, c** MDA-MB-231 cells knock-in expressing wild-type ME2fl (WT), ME2flS9A or ME2flS9D were subcutaneously injected into nude mice, and tumors were photographed 2 weeks after injection (left) and weighed (right) (**a**) (*n* = 6 mice for each group). The tumors were also lysed for the measurements of lactate (left) and pyruvate (right) abundance (**c**). *n* = 3 mice for each group. Data are means ± SD, two-tailed Student's *t* test. **b** Each nude mouse was injected intravenously with 10 nmol 2-DG-750. 2 weeks after subcutaneous injection of MDA-MB-231 cells that had knock-in expression of wild-type ME2fl (WT), ME2flS9A or ME2flS9D. Glycolytic rate in tumors was probed with fluorescence imaging of 2-DG absorption (left), and total radiation efficiency was calculated (right) (*n* = 3 mice for each group). Scale bars, 1 cm. Data are means ± SD, two-tailed Student's *t* test. **d** Expression of PTEN and AKT in prostate tissues of wild type and *Pten*-conditional knockout mice was determined by western blot analysis. Representative results are shown from three independent experiments. **e**, **f** Prostate tissues of the wild-type and *Pten*-condition knockout mice were analyzed by HE or IHC staining with anti-pAKT-S473 and anti-ME2 antibodies (**e**), as well as anti-ME2fl and anti-pME2flS9 antibodies (**f**) respectively. Scale bars, 5 mm (bright field), 200 μm (left) and 20 μm (enlarged image, right). Representative results are shown from three independent experiments. **g** Glycolytic rate was measured by intravenously injecting 10 nmol 2-DG-750 into 10-week-old *Pten*-conditional knockout or control mice for 3 h (*n* = 3 mice per group). Fluorescence of 2-DG-750 was photographed (left), and total radiation efficiency was calculated (right). Scale bars, 1 cm. Data are means ± SD, two-tailed Student's *t* test. **h** The activity of the indicated glycolytic enzymes in prostates of *Pten*-conditional knockout or control mice was measured (*n* = 3 mice for each group). Data are means ± SD, two-tailed Student's *t* test.

10 min at 4 °C. AKT1 was immunoprecipitated with anti-Flag M2 affinity gel at 4 °C for 12 h. The on-beads immunoprecipitants were washed 3 times with IP lysis buffer and eluted by 3Flag peptide in a shaker at 100 × *g* for 2 h at 4 °C to obtain recombinant Flag-AKT1 proteins. pGEX-6P-1-5'GST-Vector, pGEX-6P-1-5'GST-ME2flWT, pGEX-6P-1-5'GST-ME2flS9A or pET21b-ME2flWT-3'His, pET21b-ME2flS9A-3'His plasmids were transformed into BL21/DE3. After inducing by IPTG, bacterially purified ME2fl (5'GST-ME2flWT and ME2flWT-3'His) or mutants (5'GST-ME2flS9A and ME2flS9A-3'His) proteins were incubated with Flag-tagged AKT1 in kinase reaction buffer (10 mM Tris-HCl, pH 7.5, 10 mM MgCl$_2$, 0.1 mM EDTA and 1 mM DTT) in the presence of 1 mM ATP and at 30 °C for 30 min. The reaction was then stopped by adding 5 × SDS loading buffer and boiled at 98 °C for 10 min. The samples were analyzed by immunoblotting.

### Enzymatic activity measurements

ME2 enzymatic activity. Cells were lysed in cell fraction buffer (20 mM HEPES-KOH, pH 7.5, 10 mM KCl, 1.5 mM MgCl$_2$, 1 mM sodium EDTA buffer, 1 mM sodium EGTA buffer, 1 mM dithiothreitol, and protease inhibitor cocktail) in the presence of 250 mM sucrose. The mitochondrial fraction or purified ME2 protein concentration was determined by BCA and protein samples were added to enzyme reaction mix containing 50 mM Tris-HCl pH 7.4, 10 mM MgCl$_2$, 0.3 mM NAD$^+$, and 3.3 mM L-malic acid. The reactions were measured by absorbance at 340 nm every 20 s for up to 15 min. The mix without L-malic acid was defined as background control. Enzymatic activity was determined by subtracting the activity of the background control to each sample. The absorb changes were normalized to protein concentration.

PFK enzymatic activity. The PFK enzymatic activity was determined as previously described. Cells were lysed in IP lysis buffer and protein concentration was determined by BCA. The reaction started by adding fresh cell lysates into the PFK enzymatic assay buffer (50 mM Tris-HCl pH 7.5, 100 mM KCl, 5 mM MgCl$_2$, 1 mM ATP, 1 mM NADH, 5 mM Na$_2$HPO$_4$, 0.1 mM AMP, 1 mM NH$_4$Cl, 5 mM fructose-6-phosphate, 5 units triose phosphate isomerase, 1 unit aldolase., and 1 unit α-glycerophosphate dehydrogenase). The decrease in absorbance at 340 nm was measured every 20 s for up to 60 min.

GAPDH enzymatic activity. Cells were lysed in IP lysis buffer and protein concentration was determined by BCA. The GAPDH enzymatic activity was measured in a GAPDH enzymatic assay buffer (25 mM Triethanolamine pH 7.5, 25 mM sodium phosphate, 0.2 mM EDTA, 5 mM NAD$^+$, 5 mM glyceraldehyde-3-phosphate). The reactions were measured by absorbance at 340 nm every 20 s for up to 30 min.

PKM enzymatic activity. Pyruvate kinase activity was measured by an LDH-coupled enzyme assay. Briefly, cells were lysed in IP lysis buffer and protein concentration was determined by BCA. PKM enzymatic activity was measured in PKM enzymatic assay buffer (50 mM Tris-HCl pH 7.4, 100 mM KCl, 5 mM MgCl$_2$, 1 mM ADP, 0.5 mM Phosphoenol pyruvate, 1 mM NADH and 8 units of lactate dehydrogenase). The decrease in absorbance at 340 nm from the oxidation of NADH to NAD$^+$ was recorded.

LDH enzymatic activity. Cells were lysed and protein concentration was determined as previously described. The LDH enzymatic assay was performed in LDH reaction buffer (50 mM Tris-HCl, pH 7.6, 2 mM sodium pyruvate and 1 mM NADH). The cell lysates were then added into the reaction buffer and the decrease in absorbance at 340 nm was recorded every 20 s for up to 10 min.

### Soft agar and Xenograft assays

MDA-MB-231 cells were cultured to 80%‑90% confluence. DMEM supplemented with 0.6% agar was plated in a 12-well cell culture plate as agarose base. Cells were trypsinized and suspended DMEM supplemented with 20% FBS and 0.3% agarose and plated on 0.6% agarose base previously described (1000 cells per well). Cells were then cultured in a 5% CO$_2$ incubator at 37 °C for 2 weeks. Colonies were fixed with 10% formaldehyde and stained with 0.05% crystal violet till colonies turned into blue.

For the mouse xenograft experiment, 4 × 10$^6$ cells were injected subcutaneously into the flanks of 4 to 5-week-old athymic Balb-c nu/nu male mice (Huafukang, Beijing). 2 weeks after implantation, tumors were photographed and weighed. All the procedures performed in this study were approved by the Tsinghua University Animal Care and Use Committee (TUACUC). All the tumors' sizes didn't exceed the TUACUC-approved maximum size (10% of mouse weight, 1.8–2.0 g usually), and maximum permitted volume (1000 mm$^3$ usually). All animal were kept according to guidelines and regulations approved by the Tsinghua University Animal Care and Use Committee.

### Mass spectrometry analysis

For identification of proteins that may interact with phosphorylated ME2fl, HEK293T cells were transfected with 3'Flag tagged phospho-mimic ME2flS9D (pRK5-ME2flS9D-3'Flag) with Lipofectamine 2000. 36 h after transfection, cells were lysed with IP lysis buffer supplemented with protease and phosphatase inhibitors. Proteins were immunoprecipitated with anti-Flag M2 Affinity gel. The immunoprecipitants were eluted and separated using SDS-PAGE gel and stained by Coomassie blue. Gel bands of interest were excised and digested in gel with modified sequencing-grade trypsin (Promega) in 50 mM ammonium bicarbonate buffer overnight at 37 °C. The peptides were extracted twice with 0.1% trifluoroacetic acid in 50% acetonitrile aqueous solution for 30 min. The extracts were centrifuged and dried in a speedVac system. Peptides extracted were dissolved in 0.1% trifluoroacetic acid and analyzed by a high-sensitivity LC−MS/MS with an Orbitrap Elite mass spectrometer (ThermoFisher Scientific). Proteins identification results were analyzed by searching the fragment spectra against the UniProt protein database (EMBL-EBI) using the Mascot search engine (v.2.3; Matrix Science) with the Proteome Discoverer software program (v.1.4; ThermoFisher Scientific).

For identification of phosphorylation site(s), HEK293T cells were transfected 5'Flag tagged ME2fl (pRK5-5'Flag-ME2fl) together with HA tagged control vector or AKT1. 36 h after transfection, cells were harvested and lysed in IP Lysis Buffer with phosphatase and protease

inhibitors. After immunoprecipitation by Flag M2 Affinity gel, phosphorylated ME2fl proteins were separated using SDS-PAGE gel. The proteins were excised and digested in gel with modified sequencing-grade trypsin. Digested peptides were separated by a 60-min gradient elution at a flow rate of 0.3 μL·min⁻¹ with the Dionex/Thermo UltiMate 3000 HPLC System that was directly interfaced with Orbitrap Fusion mass spectrometer (ThermoFisher Scientific). The analytical column was a homemade fused silica capillary column (75 μm ID, 150 mm length) packed with C-18 resin (300 A, 5 μm). Mobile phase A consisted of 0.1% formic acid, and mobile phase B consisted of 80% acetonitrile and 0.08% formic acid. The Orbitrap Fusion mass spectrometer was operated in the data-dependent acquisition mode using Xcalibur 4.0.27.10 software and there is a single full-scan mass spectrum in the Orbitrap (300–1800 $m/z$, 17,500 resolution) followed by four targeted tandem mass spectrometry scans at 30% normalized collision energy. Static peptide modifications included carbamidomethylation (C), dynamic oxidation (M) and phosphorylation (S, T and Y). One trypsin missed cleavage was allowed. Precursor tolerance and ion fragment tolerance were set at 20 ppm and 0.05 Da, respectively. Confidence levels were set to 1% FDR (high confidence) and 5% FDR (middle confidence).

### Edman N terminal sequencing

HEK293T cells transfected with pRK5-5'Flag-ME2fl or pRK5-ME2fl-3'Flag were lysed with IP lysis buffer and immunoprecipitated with anti-Flag M2 affinity gel (A2220, Sigma) at 4 °C overnight. The immunoprecipitations were washed three times with IP lysis buffer and eluted with 0.1 M glycine (pH 2.5) at 100 × $g$ for 5–10 min followed by neutralization with 100 mM Tris buffer for immunoblotting analysis. After boiling in SDS-loading buffer at 100 °C for 15 min, samples were separated on an SDS–PAGE gel before being transferred to a polyvinylidene fluoride (PVDF) membrane in 3-clohexylaminopropanesulfonic acid (CAPS) buffer at pH 11.0. Proteins on the membrane were located by staining with 1% Coomassie blue for 1 min, followed by destaining in 50% methanol. The N-terminal fragments were sequenced by Edman degradation peptide sequencing (PPSQ-33A, Shimadzu) and five cycles were set. The amino acid sequences of samples were determined by comparing chromatograms and identifying the phenylthiohydantoin amino acids that had the greatest increase in abundance. The N-terminal sequence of protein was read up to 10 amino-acid residues.

### Protein N terminal sequencing by mass spectrometry

HEK293T cells were transfected with pRK5-5'Flag-ME2fl or pRK5-ME2fl-3'Flag and proteins were purified as described above. Samples were separated on an SDS–PAGE gel and transferred to PVDF membrane in CAPS buffer. Protein bands were excised after being stained with 1% Coomassie blue. Proteins were digested with sequencing-grade trypsin (Promega) and incubated overnight at 37 °C. Peptides were analyzed by an Orbitrap Fusion Lumos spectrometer (Thermo Fisher Scientific).

### Mass spectrometry-based metabolic flux analysis

HepG2 or PC9 cells expressing shRNA targeting PTEN or control shRNA, or HCT116 cells stably expressing ME2fl (pCDH-ME2fl -3'Flag), ME2flS9A (pCDH-ME2flS9A-3'Flag), ME2flS9D (pCDH-ME2flS9D-3'Flag), or vector control (pCDH-Flag Vec) were seeded into 6 cm cell culture dishes with a confluence of 80%.

For glycolytic flux measurement, cells were cultured in DMEM medium (11966-025, Thermo Fisher Scientific) containing 10 mM [U-¹³C₆]glucose and dialyzed FBS(30067334, Thermo Fisher Scientific). For TCA cycle flux measurement, cells were cultured in DMEM medium (31053-028, Thermo Fisher Scientific) containing 4 mM [U-¹³C₅]glutamine and dialyzed FBS. After 6 h, cells were collected and metabolites were extracted with cold 80% methanol (v/v) by thorough mixing. Extracts were centrifuged at 14,000 × $g$ for 10 min and the supernatant was evaporated to dryness under vacuum in a SpeedVac evaporator.

Flux assay was performed on TSQ Quantiva Triple Quadrupole mass spectrometer (Thermo Fisher Scientific) with positive/negative ion switching. Mobile phase A was prepared by adding 2.376 ml tributylamine and 0.858 ml acetic acid to HPLC-grade water, then adding HPLC-grade water to 1 L volume. Mobile phase B was HPLC-grade methanol. Synergi Hydro-RP 100 A column was used for polar metabolites separation with column temperature at 35 °C. The measured mass isotopomer distributions were corrected by natural abundances.

### Hematoxylin and Eosin (HE) and Immunohistochemistry (IHC) staining

The wild type *Pb-Cre⁻;Pten^{L/L}* and *Pten* condition knock-out *Pb-Cre⁺;Pten^{L/L}* male mice were sacrificed at 10-week old and the prostate tissues were fixed in 4% PFA overnight and then dehydrated in increasing concentrations of ethanol, followed by clearing of ethanol by xylene (70% ethanol for 2 times, 1 h each; 80% ethanol, 1 h; 95% ethanol, 1 h; 100% ethanol for 3 times, 1.5 h each; xylene for 3 times; 1.5 h each). The tissues were finally embedded in paraffin wax in cassettes for facilitation of tissue sectioning. Paraffin blocks were trimmed, cut at 5 μm thick and the paraffin ribbons were placed in water bath at 40 °C. The sections were mounted onto slides, dried for 30 min at room temperature and baked in an incubator at 37 °C overnight.

Before HE staining, sections were dewaxed with the following steps: xylenes, 2 times, 2 min each; 100% ethanol, 2 times, 2 min each; 95% ethanol, 2 times, 2 min each; 80% ethanol, 2 min; 75% ethanol, 2 min; 50% ethanol, 2 min. The dewaxed samples were brought to distilled water and nuclei were stained with the haematoxylin for 5 min. Afterwards, the sections were rinsed in running tap water, differentiated with 0.3% acid alcohol, rinsed in tap water, and stained with eosin for 2 min. The tissues were dehydrated in alcohol (70%, 95%, 100%), 3 min each and cleared with xylene or and mounted with resinene.

For Immunohistochemistry, tissues were fixed in 4% PFA, dehydrated, and dewaxed as described above. The slides were incubated in citrate buffer at 95 °C for 20 min for antigen retrieval and in 3% $H_2O_2$ for 10 min, washed in distilled $H_2O$ two times for 5 min each, blocked with goat serum and then incubated overnight at 4 °C with the primary antibodies including anti-mouse ME2fl (1:100), anti-ME2 (1:200), anti-p-AKT S473 (1:100). After washing, tissues were incubated with streptavidin–horseradish peroxidase conjugated goat anti-mouse/rabbit secondary antibodies for 1 h at room temperature. After 3 times washing, DAB solution was added and the slides were counterstained with haematoxylin for 5 min at room temperature. The sections were washed 3 times and dehydrated as the following steps: 95% ethanol 2 times for 10 s each, 100% ethanol 2 times for 10 s each, xylene 2 times for 10 s each. Sections were mounted with resinene and coverslips. The tissue sections were scanned using Pannoramic SCAN (3DHISTECH Ltd.).

### Extracellular acidification rate (ECAR) and Oxygen consumption rate (OCR) analysis

The ECAR and OCR were determined with a Seahorse XFe96 extracellular flux analyzer (Agilent Technologies) as described in the manufacture's protocol. $1 × 10^4$ cells per well were seeded in 96-well cell culture plates (Agilent Technologies) in DMEM with 10% FBS and incubated at 37 °C overnight in a 5% $CO_2$ incubator. For ECAR analysis, cell culture medium was replaced with bicarbonate-free base ECAR medium (pH 7.4, containing 4 mM glutamine) followed by incubation at 37 °C in a non-$CO_2$ incubator for 1 h to equilibrate the $CO_2$ level in the atmosphere. ECAR were measured by the treating cells with 25 mM glucose, 1 μM oligomycin and 100 mM 2-DG. For OCR analysis, the cell culture medium was replaced with bicarbonate-free base OCR medium (pH 7.4, containing 25 mM glucose, 4 mM L-glutamine, and 2 mM sodium pyruvate) followed by incubation at 37 °C in a non-$CO_2$ incubator for 1 h to equilibrate the $CO_2$ level in the atmosphere. Cells were

treated with 1 μM oligomycin, 1 μM FCCP and 2.5 μM Antimycin/Rotenone for OCR measurement. Each measurement cycle consisted of 3 min of mixing, 3 min of waiting, and 4 min of measuring. The ECAR and OCR values were normalized to protein concentrations and analyzed using WAVE software (Agilent Technologies).

## Metabolite analysis

HCT116 cells were grown in 10 cm plates to 80% confluence, the culture medium was rapidly aspirated and cells were washed with cold PBS on ice. 1 mL extraction solvent (80% methanol/water) cooled to −80 °C was added to each well, and cells were then scraped into the extraction solvent and transferred to 1.5 mL tubes on dry ice. Cells were vortexed for 30 s and centrifuged at 13,800 × g at 4 °C for 10 min. For tissue metabolite analysis, 50 mg samples were lysed with 1 mL extraction solvent followed by centrifugation at 13,800 × g at 4 °C for 10 min. Either cell or tissue supernatant was transferred to new 1.5 mL tubes, evaporated in a Speed Vacuum and stored at −80 °C until they were run on the mass spectrometry. Metabolites extracted were analyzed by LC-MS/MS.

## ROS measurement

Cells were incubated at 37 °C for 30 min in PBS containing 10 μM 2′,7′-dichlorodihydrofluorescein diacetate (H2-DCFDA). After incubation, cells were digested and re-suspended in PBS. Fluorescence was measured using a FACScan Flow Cytometer (Becton Dickinson).

## Lentivirus Infection

pCDH-hygro-ME2fl wild-type (pCDH-ME2flWT-3′Flag) and ME2fl mutant constructs (pCDH-ME2flS9A-3′Flag and pCDH-ME2flS9D-3′Flag) were constructed as previously described. The oligonucleotides targeting PTEN (5′-CCGGAGGCGCTATGTGTATTATTATCTCGAGATAATAATACACATAGCGCCTTTTTT-3′) were annealed into a pLKO.1 shRNA vector. Lentiviruses were produced by transfecting HEK293T cells with 4 μg recombinant plasmids and packing plasmids (2 μg VSVG and 3 μg PSPAX2) using Lipofectamine 2000 transfection regent (ThermoFisher Scientific) according to the manufacturer's instructions. Viral supernatants were collected at 48–72 h post-transfection, passed through a 0.45 μm filter, diluted in fresh medium containing 8 μg/mL polybrene and used to infect the HCT116 cells and HepG2 cells at 80% confluence. Cells were selected with 50 μg/mL hygromycin for pCDH backbone virus infection or 1 μg/mL puromycin for shRNA infection.

## In vivo imaging

The 10-week-old *Pb-Cre⁻;Pten^{L/L}* and *Pb-Cre⁺;Pten^{L/L}* mice were intravenously injected with 10 nmol of 2-DG-750 probe for 3 h and imaged with IVIS Spectrum (Ex745 nm/Em820 nm, PerkinElmer). The probe was warmed up to room temperature before injection in an animal.

## Electron microscopic immunogold staining

Cells were seeded in 6 cm cell culture plate and cultured overnight to 80% confluence. The cells were fixed with 2% formaldehyde diluted by 0.1 M PB buffer (containing 0.07541 M Na₂HPO₄ and 0.02459 M NaH₂PO₄, pH 7.4) for 2 h at room temperature followed by fixation in 1% formaldehyde at 4 °C. Cell samples were rinsed 3 times (10 min each) in 0.1 M phosphate buffer and quenching the aldehyde group with 0.1% fresh-made NaBH₄ for 5 min. For dehydration, the cells were treated with graded ethanol series as described above. After embedded in LR white, ultrathin sections were prepared on a Leica UCT ultramicrotome and collected on nickel grids.

For antibody labeling, the grids were placed on the surfaces of 20% H₂O₂ droplets for 20 min and 2% BSA droplets for 30 min at room temperature. The grids were then transferred to the surfaces of primary antibody droplets (anti-ME2, 1:20 dilution in 2% BSA) and incubated overnight at 4 °C. Non-specific bound ME2fl antibody was washed with 20 μL droplets of PB buffer for 5 times and then the grids were transferred to droplets of 6 nm Colloidal Gold AffiniPure Goat Anti-Rabbit IgG (H + L) (Jackson ImmunoResearch Laboratories, 111-195-144; 1:50 dilution in 2% BSA) for 2 h at room temperature. Unbound gold conjugate was washed by transferring the grids to a series of buffer (25 μL droplets of PB for 3 times, 5 min each; 25 μL droplets of 1% glutaraldehyde in PB for once, 10 min; 25 μL droplets of distilled water for 5 times, 1 min each). After washing, the grids were stained in uranyl acetate and examined with an electron microscope (HT7700 120 kV Compact-Digital Biological Transmission Electron Microscope, Hitachi)

## Structured Illumination super-resolution Microscope (SIM)

SIM was performed as described in immunofluorescence with some modifications. Briefly, cells were seeded in an 8-well chamber for immunofluorescence and cultured to 50–60% confluence. Cells were washed with PBS once and fixed by PFA buffer (3% PFA and 0.1% glutaraldehyde diluted in PBS) for 10 min at room temperature. After fixation, cells were incubated with 0.1% NaBH₄ in distilled water, washed with PBS for 3 times 5 min each, and blocked with BSA buffer (3% BSA and 0.2% TritonX-100, diluted in PBS) for 1 h at room temperature. Primary antibody (anti-ME2, CST, 35939; 1:400 diluted in BSA buffer) was added into the chamber for overnight incubation at 4 °C. After washing for 3 times with Wash buffer (0.2% BSA and 0.02% TritonX-100 in PBS) 5 min each, cells in the chamber were stained with secondary antibody (ThermoFisher Scientific, A11008; 1:1000) for 1 h at room temperature. The cell samples were finally fixed in PFA buffer for 10 min at room temperature and stored in PBS at 4 °C. The images were scanned by a super resolution microscope (N-SIM S, Nikon)

## ME2fl knock-in cell lines

Genomic mutations were performed using the CRISPR–Cas9 system. Single-guide RNA (sgRNA) targeting the genomic area adjacent to mutation site (ME2flS9A and ME2flS9D) was designed using the CRISPR design tool (http://crispr.mit.edu/). The annealed oligonucleotides were inserted into a PX330-GFP vector digested with BbsI restriction enzyme. Both upstream and downstream of the donor nucleotides were 350 bp away from mutation site and the donor oligonucleotides were inserted into a pUC57 vector. SgRNA and donor plasmids were co-transfected with a 1:1 ratio (1 μg sgRNA and 1 μg donor plasmids) into MDA-MB-231 cells at 70% confluence in a 6 cm cell culture plate. 24 h after transfection, cells were trypinzed and GFP positive cells were sorted by FACScan Flow Cytometer (Becton Dickinson) for GFP positive single cell and seeded in 96-well plates. The sgRNA sequence targeting ME2fl was 5′-TGAAAGAAAAGATGTTGTCC-3′.

The donor sequence for ME2flS9A was: 5′-ACAAGTGGGAGATGCATACCTTGCCACACTTGTGTGACTAGAAAGAGGTTAAATGTTTTTTGCAGTGTCTGATACATAGTAGGTGGTACTTAAAGAGTAGCAGTGATTATTTCCCAAGAACATAAGAGTGTTATCAGAAAAAGTAAGTTTTCACCTTCGCCTCTTGTTACCTTTTTAAAACTGTTGCAGCCTGTGGAAACATAAATGTGTTTTGAAAACTGATACCCGATGGACATGAAGGCCTATAATATGATTCTCTTCAGTGGTTTATTTGCTTTGTTTTTCTCAGGTGAAAGAAAAGATGTTGTCCCGGTTAAGAGTAGTTgCCACCACTTGTACTTTGGCATGTCGACATTTGCACATAAAAGAAAAAGGCAAGCCACTTAGCTGAACCCAAGAACAAACAAGGTTAGTAACATTAATATCAATGTACATTTTCTCTCTTCTTATTAAAGTTTGATATATTTAAGACTGATAAGGTGTGACATATTTTTGTAGCTTTATGCTAAGGTTGTTCAGAGATAGTTGTTTACATGAAATTTTCACGCTTCTCCACATGGCAAAATCGTGGTTGCTTTATACCTCCTCAATTTCATCCTCATGACTTTTACC-3′;

The donor sequence for ME2flS9D was: 5′-ACAAGTGGGAGATGCATACCTTGCCACACTTGTGTGACTAGAAAGAGGTTAAATGTTTTTTGCAGTGTCTGATACATAGTAGGTGGTACTTAAAGAGTAGCAGTGATTATTTCCCAAGAACATAAGAGTGTTATCAGAAAAAGTAAGTTTTCACCTTCGCCTCTTGTTACCTTTTTAAAACTGTTGCAGCCTGTGGAA

ACATAAATGTGTTTTGAAAACTGATACCCGATGGACATGAAGGCCT
ATAATATGATTCTCTTCAGTGGTTTATTTGCTTTGTTTTTCTCAGGTG
AAAGAAAAGATGTTGTCCCGGTTAAGAGTAGTTgaCACCCACTTGTAC
TTTGGCATGTCGACATTTGCACATAAAAGAAAAAGGCAAGCCACTTAT
GCTGAACCCAAGAACAAACAAGGTTAGTAACATTAATATCAATGTACA
TTTTCTCTCTTCTTATTAAAGTTTGATATATTTAAGACTGATAAGGTG
TGACATATTTTTGTAGCTTTATGCTAAGGTTGTTCAGAGATAGTTGTT
TACATGAAATTTTCACGCTTCTCCACATGGCAAAATCGTGGTTGCTT
TATACCTCCTCAATTTCATCCTCATGACTTTTACC-3′

The lower-case letters in the donor sequences indicated the mutated nucleotides that will replace the endogenous nucleotides in the genomic DNA.

Genotyping was performed by sequencing PCR products amplified from the following primers. The forward primer sequence was 5′-CACCTTCGCCTCTTGTTACC-3′ and the reverse primer sequence was 5′-CTACAAAAATATGTCACACCTTATC-3′.

## Sucrose density-gradient centrifugation

3%–55% continuous sucrose gradient was prepared in a centrifuge buffer containing 50 mM Tris-HCl (pH 8.0), 50 mM HEPES-KOH and 2 mM EDTA in ultracentrifuge tubes. 0.4 mL of a 1:1 mixture of cell lysate with 3% sucrose solution was loaded on a total of 2 mL gradient solution. The samples were centrifugated at $214,000 \times g$ for 4.5 h (TSL-55 rotor, Beckman ultracentrifuge). Twelve equal volume fractions were collected and subjected to Western Blot analysis after centrifugation.

## In vitro glycolysis analysis

Flag-tagged PGI, PFKL, ALDOA, TPI, GAPDH, PGK, PGAM, ENO, PKM2, and LDHA proteins were purified from HEK293T cells and incubated in the glycolytic assay buffer containing 50 mM Tris-HCl, 15 mM $MgCl_2$, 2 mM $K_3PO_4$, 1 mM G6P, 1 mM ATP, 1 mM ADP, 1 mM $NAD^+$, 1 mM NADH in the presence of bacterially purified control GST-Vector or GST-ME2fl. After incubation for 1 h at 37 °C, reactions were stopped by adding a 1:1 mix of methanol with acetonitrile (ACN) and the metabolites were analyzed by LCMS.

## Statistics and reproducibility

Sample sizes were chosen based on previous publications and were sufficient for statistical analysis. No data were excluded from analysis. Animals were randomly assigned to each group. Randomization was not required for biochemical and in vitro experiments. Cellular and biochemical experiments were not blinded. All statistical data are presented as means ± SD. The statistical $P$ values were obtained using the GraphPad Prism software 9.0 (GraphPad Software, Inc. USA). Tests performed with $P < 0.05$ were considered statistically significant. Statistical significance is shown as $*P < 0.05$, $**P < 0.01$, $***P < 0.001$, $***P < 0.0001$.

## Reporting summary

Further information on research design is available in the Nature Portfolio Reporting Summary linked to this article.

## Data availability

The proteomics and mass spectrometry data generated in this study have been deposited in the ProteomeXchange database under accession code PXD040380 and PXD040377, respectively. The metabolomics data used in this study have been deposited in the MetaboLights database under accession code MTBLS8218. The processed metabolomics data in this study are provided in the Source Data file. Source data are provided with this paper.

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

## Acknowledgements

We thank Drs. Li Yu, Wei Wu, Mo Chen, Deng Pan, Xin Lin, and Yiguo Wang for reagents and/or helpful comments. We thank all the Jiang laboratory members for technical assistance and/or discussion. We thank Xueying Wang, Lina Xu and Xiaohui Liu for helping with the LC–MS/MS experiments. This work was supported by the National Natural Science Foundation of China (81930082, 82125030 and 82341022) to P.J., and the National Key Research and Development Program of China (2022YFA0806302, 2019YFA0802600), CAMS Innovation Fund for Medical Sciences (CIFMS) (2021-I2M-1-016), Haihe Laboratory of Cell Ecosystem Innovation Fund (22HHXBSS00011), National Natural Science Foundation of China (81672766), CAMS Basic Research Fund (2019-RC-HL-007), and State Key Laboratory Special Fund (2060204) to W.D. The working model was created with BioRender.com.

## Author contributions

T.C. and S.X. performed all the experiments. J.C. provided technical assistance. T.C. and P.J. designed the experiments. Q.Z. provided technical assistance. H.W. and W.D. provided reagents and constructive discussion. T.C. collected and analyzed the data. P.J. supervised the project, interpreted the data and wrote the manuscript.

## Competing interests

The authors declare no competing interests.
