## [Peer Review File · Nature Communications]

Reviewers' Comments:

Reviewer #1:

Remarks to the Author:

This manuscript presents a novel hypothesis that Malic enzyme 2c (ME2c) is phosphorylated by AKT and is localized in cytoplasm, thereby promoting glycolysis. The ME2 localization data and AKT driven phosphorylation data is solid and strong. However, metabolomics and metabolite profiling data is pretty weak. Only Seahorse is used to indicate increased glycolysis. Here are my recommendations to improve the authors' claims:

1. Authors should perform ¹³C glucose and glutamine tracing in PTEN normal, Pten null, as well as in HepG2 and PC9 cells. Based on the isotopologue distributions author can make claim about increased glycolysis or TCA cycle activity.
2. Further, ¹³C glucose and glutamine tracing should also be performed with HCT116 cells stably expressing wild-type ME2c (pCDH-ME2c-3'Flag), ME2cS9A (pCDH-ME2cS9A-3'Flag), ME2cS9D (pCDH-ME2cS9D-3'Flag), or vector control (pCDH Flag Vec).
3. Further, role of ME1 and ME3 is not discussed in this regard. There is quite possible that metabolic compensation exists between different isoforms. How, will authors suggest that ultimate glycolytic activity is not affected by ME1.
4. NADPH data is less convincing. Authors should measure through Mass Spec, NAD, NADH, and NADPH to clearly ascertain the role of mitochondrial ME2 in their model.
5. Will TGF-beta in the media regulate ME activity?
6. Is there changes in ROS in low pH cultures?
7. Does cytosolic ME2 integrates with mitochondrial IDH2 and regulates growth and metastasis as shown in "Cytosolic ME1 integrated with mitochondrial IDH2 supports tumor growth and metastasis".

Reviewer #2:

Remarks to the Author:

In this work, Professor Jiang and colleagues have identified a novel phospho-regulatory mechanism by which growth factor signaling can alter the metabolism of cancer cells. This began with the observation that the metabolic enzyme malic enzyme 2 (ME2), normally mitochondrial, was localized to the cytosol specifically in PTEN null cells. The authors then demonstrated an elegant mechanism to explain this: the growth factor signaling kinase AKT targets and phosphorylates ME2 at its mitochondrial leader sequence (Ser9 – which they identified by mass spec), which blocks its entry into the mitochondrial matrix. Once phosphorylated and trapped in the cytosol, ME2 has higher catalytic activity and functions as a scaffold for multiple glycolytic enzymes, thereby increasing the glycolytic flux to promote anabolic processes and cell growth.

Overall, this solid study reveals a novel and biochemically plausible regulatory mechanism for coordinating growth factor signaling and metabolism. I can imagine this being of interest in cancer and other biological fields as this work involves the phosphoregulation of a widely expressed metabolic enzyme (ME2) by a major signaling kinase (AKT).

Below, I have outlined points for the authors to address.

1. The phosphorylation site (Ser9) on ME2 is arginine-rich and matches the known substrate-targeting motif of AKT. However, it also conforms quite well to S6K's motif (a major downstream kinase of AKT), and thus S6K might cooperate with AKT in regulating the localization of ME2. Can the authors examine the effects of S6K-specific inhibitors on Ser9 phosphorylation in the insulin treatment setting (or alternatively in other related experimental settings)?

2. The authors show that cells expressing the S9E mutant (that is partially retained in the cytosol) are more proliferative. Two underlying mechanisms are proposed to explain this: a) a catalytic-independent mechanism where ME2 promotes the assembly of glycolytic enzymes to enhance glycolytic flux and b) a catalytic-dependent mechanism where phosphorylation enhances its enzymatic activity and NADPH generation in the cytosol (and correspondingly, its absence of activity in the mitochondria and inability to contribute NADH).

Would introducing catalytically inactivating mutations (for example, D279A, G446D, or R670Q (PMID: 26008970)) into ME2 S9E impact its effects on cell proliferation? These questions may be satisfactorily addressed by shorter time frame experiments, such as colony formation assays with ME2 S9E catalytic dead knock-in expression cell lines (ED Fig 10a).

3. The leader sequence (first 18 amino acids) on ME2 contains two cysteines (Cys12 and Cys16) that are adjacent to the identified phosphorylation site (Ser9) and appear well conserved. It is interesting to speculate that the cellular redox state might regulate the mitochondrial entry of ME2 through the oxidation of these cysteines, which would introduce negative charge similar to phosphorylated residues. This may be a mechanism for countering redox stress and might be worth commenting on in the discussion section, as it would be in keeping with the impressive reduction of ROS seen in Fig 5i when ME2 is retained in the cytosol.

4. Introduction/background section: This would benefit from a brief background on ME2 and its canonical roles, as it is the paper's primary focus and is less widely known than AKT (which is discussed at length in the intro).

5. Figure 1a-In the legend, please clarify what the colors of the arrows depict. (e.g., black arrows represent mitochondrial localization, and red arrows represent cytosolic or non-mitochondrial localization.

Figure 1c – same as 1a regarding white and green arrows.

6. The label 'ME2c' (where c refers to cytoplasmic) is potentially misleading because it implies that the protein is cytosolic when referring to the full-length form. I understand this label was coined to distinguish the full-length form from the cleaved form 'ME2m' that had entered the mitochondria, as illustrated in Fig 2b. However, using 'ME2c' outside these contexts is potentially confusing (e.g., labeling expression constructs as ME2c, showing confocal images of the mitochondrial localization of ME2c). Unless the authors can provide a compelling reason for keeping it, I feel that the 'c' should be dropped from the manuscript, and only 'ME2m' needs to be specified when appropriate.

Reviewer #3:

Remarks to the Author:

The paper presents an intriguing discovery of a previously unknown mechanism by which AKT1 regulates glycolysis and the mitochondrial TCA cycle through ME2 phosphorylation. The authors provide a detailed account of the interaction between ME2c and key glycolytic enzymes, enhancing our understanding of how tumor cells coordinate glycolysis in response to growth stimuli.

The authors have employed a variety of experimental techniques, including immunoprecipitation analysis, confocal microscopy, and in vivo tumor models, to explore the role of ME2c phosphorylation in tumorigenesis. A more detailed description of the experimental methods and controls used would be helpful to allow readers to better evaluate the validity of the findings.

Comments:

1. The authors should discuss potential limitations of the study and any assumptions made in interpreting the data. This would provide a more balanced view of the results and their implications.

2. The discussion section could be expanded to include a broader perspective on the role of AKT1

in metabolic reprogramming and its implications for cancer biology. Additionally, the authors should consider possible future research directions, including the investigation of other molecular mechanisms that may be involved in ME2c phosphorylation and glycolytic regulation.

In the conclusion, the authors should provide a clear summary of the main findings, emphasizing the novelty of their work and its potential implications for the understanding of tumor cell metabolism and treatment strategies. Better draw an illustrative picture to show molecular mechanisms by end.

Accept after major revisions

Point-by-point response to reviewers' comments

Reviewer #1 - mitochondrial metabolism, Metabolomics, mass-spec - (Remarks to the Author):

This manuscript presents a novel hypothesis that Malic enzyme 2c (ME2c) is phosphorylated by AKT and is localized in cytoplasm, thereby promoting glycolysis. The ME2 localization data and AKT driven phosphorylation data is solid and strong. However, metabolomics and metabolite profiling data is pretty weak. Only Seahorse is used to indicate increased glycolysis. Here are my recommendations to improve the authors' claims:

We very much thank the referee for the positive and constructive comments. As detailed below, we have performed numerous experiments to address these comments, which we believe improves the manuscript greatly. Please note that new figures generated during the revision are indicated in red color.

1. Authors should perform ¹³C glucose and glutamine tracing in PTEN normal, Pten null, as well as in HepG2 and PC9 cells. Based on the isotopologue distributions author can make claim about increased glycolysis or TCA cycle activity.

We appreciate the comment by the referee. As suggested, we have performed ¹³C-isotope tracing analysis using ¹³C-glutamine and ¹³C-glucose. Due to the heterogeneity between cell lines from different tissue sources, direct comparisons of changes in metabolism within different cell lines may not be definitive. Therefore, we directly knocked down PTEN expression in hepG2 and PC9 cells, which allowed us to compare the effects of normal and knocked-down PTEN on metabolism within these two cell lines. As shown in the **revised Extended Data Fig. 8**, PTEN knockdown resulted in increased levels of ¹³C-labelled glycolytic intermediates in HepG2 cells cultured with [U-¹³C₆]glucose. In contrast, reduced levels of ¹³C-labelled TCA cycle metabolites were observed in [U-¹³C₅]glutamine-treated HepG2 cells when PTEN was knocked down. Similar results were obtained in PC9 cells. Taken together, these data provide additional evidence that PTEN loss induces a metabolic switch from TCA cycle to glycolysis.

2. Further, ¹³C glucose and glutamine tracing should also be performed with HCT116 cells stably expressing wild-type ME2c (pCDH-ME2c-3'Flag), ME2cS9A (pCDH-ME2cS9A-3'Flag), ME2cS9D (pCDH-ME2cS9D-3'Flag), or vector control (pCDH-Flag Vec).

We thank the referee for this constructive comment. Following the referee's suggestion, we have now included data from stable isotope tracer analysis of HCT116 cells stably expressing ME2c or its mutants. Consistent with the seahorse data (**revised Fig. 5e, 5f, and Extended Data Fig. 9g**), cells expressing the pCDH-ME2cS9D-3'Flag showed increased glycolytic activity, as indicated by increased levels of ¹³C-labelled glycolysis intermediates derived from [U-¹³C₆]glucose, and correspondingly decreased TCA cycle flux (**revised extended data Fig. 10a, b**). These findings are important and further support our hypothesis that ME2c phosphorylation promotes glycolysis while reducing TCA cycle metabolism. Due to space limitations, we currently include these data in the

Extended Data figures, but if the referee prefers, certainly we can move these data to the main figures.

3. Further, role of ME1 and ME3 is not discussed in this regard. There is quite possible that metabolic compensation exists between different isoforms. How, will authors suggest that ultimate glycolytic activity is not affected by ME1.

We appreciate this thoughtful comment by the referee. Indeed, it is likely that metabolic compensation exists between different isoforms. To address this issue, we knocked down the expression of ME1 in both HCT116 and HepG2 cells expressing wild-type or mutant ME2c with the S9A or S9D mutation. Interestingly, in both cell lines, the glycolytic activity of cells expressing ME2cS9D was significantly higher than that of cells expressing wild-type ME2c or ME2cS9A, and this effect was independent of the status of ME1, as the knockdown or non-knockdown of ME1 did not affect the increased regulation of glycolytic activity by ME2cS9D (**revised Extended Data Fig. 11a, b**).

Thus, the ME2c phosphorylation-mediated increase in glycolytic activity appears to be not affected by ME1.

4. NADPH data is less convincing. Authors should measure through Mass Spec, NAD, NADH, and NADPH to clearly ascertain the role of mitochondrial ME2 in their model.

We thank the referee for this excellent comment. As suggested, we have performed mass spectrometry analysis of cellular NAD⁺, NADH and NADPH. By LC-MS analysis, we found that although wild-type ME2c and the ME2cS9A mutant endowed cells with enhanced NADH and NADPH production, ME2cS9D had a stronger effect, whereas no significant change in NAD⁺ production was observed (**revised Fig. 5g, h**). A possible explanation for these observations is that ME2cS9D not only has the enzymatic ability to synthesize NADPH, but also promotes NADH synthesis by increasing glycolytic activity.

5. Will TGF-beta in the media regulate ME activity?

We thank the referee for this comment. It has been reported that TGF- β can activate the PI3K-AKT signaling pathway through TRAF6-mediated ubiquitination of p85¹ or modulation of PTEN expression². Therefore, together with our finding that AKT phosphorylates ME2 to increase its activity, it would be expected that TGF- β treatment could increase the enzymatic activity of ME2. Indeed, treatment of HepG2 cells with TGF- β resulted in increased ME2 activity (please see **Response Fig. 1a, 1b, below**), and similar results were obtained in A549 cells (**Response Fig. 1c, 1d, below**).

Interestingly, we found that TGF- β treatment also led to an increase in ME1 activity in these cell lines (**Response Fig. 1b, 1d, below**), and this does not appear to be due to an alteration in ME1 expression, as no significant change in ME1 protein levels was observed when cells were treated with TGF- β (**Response Fig. 1e below**), suggesting an unidentified mechanism for the interaction between

TGF- β signaling and ME1. However, to fully address this issue appears to be a long-term project by itself and may be outside the scope of the current study. We hope to address it in a future work. We wish the referee would agree.

6. Is there changes in ROS in low pH cultures?

This is an excellent and very insightful comment. Indeed, AKT-mediated phosphorylation of ME2c increases glycolysis and promotes lactate production, whereas an increase in extracellular lactate is expected to decrease medium pH. To address the comment by the referee, we tested the effect of pH changes on cellular ROS in three different types of cell lines, HepG2, HCT116 and H1299. As expected, low pH was significantly correlated with increased ROS in these cells (please see **Response Fig. 2a-c below**). Because of the large amount of data that are already in the revised manuscript, these data are not included. But we will present these data if the referee prefers.

Nevertheless, we have revised the text to include a statement "In addition, low pH caused by increased lactate production may also contribute to ROS accumulation" to clarify the possibility that, in addition to NADPH reduction, low pH caused by increased glycolysis may also promote ROS production.

Response Fig. 2. Low pH increases cellular ROS. ROS levels in HepG2 cells (a), HCT116 cells (b) and H1299 cells (c) cultured for 24h in medium with different pH scales from 7.4 to pH 5.4 as indicated were determined by 2',7'-dichlorodihydrofluorescein diacetate (DCF) staining followed by FACS analysis. The data are representative of three independent experiments.

7. Does cytosolic ME2 integrates with mitochondrial IDH2 and regulates growth and metastasis as shown in "Cytosolic ME1 integrated with mitochondrial IDH2 supports tumor growth and metastasis".

We very much thank the referee for this thoughtful comment. To directly address this comment, we investigated the effect of IDH2 knockdown on cytoplasmic ME2-mediated tumor cell proliferation and metastasis, if any. Overexpression of ME2c in HCT116 and HepG2 cells increased cell proliferation in vitro, while ME2cS9D showed a stronger effect (please see **Response Fig. 3a, 3b below**). Notably, silencing of IDH2 in these cells failed to inhibit the proliferative effect of ME2c and ME2cS9D on cell proliferation (please see **Response Fig. 3a, 3b below**). Furthermore, in wound healing assays, although ME2c, especially ME2cS9D, promoted cell migration, this effect was not affected by silencing IDH2 (please see **Response Fig. 3c below**). Similar results were obtained in transwell experiments. Knockdown of IDH2 expression had no influence on ME2c or ME2cS9D-mediated enhancement of cell invasion (please see **Response Fig. 3d below**). We also wanted to know if ME2c had any effect on the expression of IDH2. Western blot results from different types of tumor cell lines with ME2 silencing showed that ME2c depletion did not alter the expression levels of IDH2 (please see **Response Fig. 3e below**).

Taken together, these data suggest that cytosolic ME2 may not integrate with mitochondrial IDH2 to regulate growth and metastasis.

Response Fig. 3. Effect of IDH2 on ME2c-regulated growth and metastasis. a, HCT116 cells (top) and HepG2 cells (bottom) stably expressing ME2c (pCDH-ME2c -3'Flag), ME2cS9D (pCDH-ME2cS9D-3'Flag), or vector control (pCDH-Flag Vec) were transfected with siRNA targeting IDH2 and control siRNA and analyzed by western blotting with indicated antibodies. **b,** Cell proliferation of HCT116 cells (left) and HepG2 cells (right) stably expressing ME2c (pCDH-ME2c -3'Flag), ME2cS9D (pCDH-ME2cS9D-3'Flag), or vector control (pCDH-Flag Vec) transfected with siRNA targeting IDH2 and control siRNA. **c,** Wound healing assay of HCT116 cells stably expressing ME2c (pCDH-ME2c-3'Flag), ME2cS9D (pCDH-ME2cS9D-3'Flag) or vector control (pCDH-Flag Vec) transfected with siRNA targeting IDH2 and control siRNA (left). Migration distances were calculated and analyzed (right). Scale bars, 100 μ m. **d.** Transwell assay HCT116 cells stably expressing ME2c (pCDH-ME2c-3'Flag), ME2cS9D (pCDH-ME2cS9D-3'Flag) or vector control (pCDH-Flag Vec) transfected with siRNA targeting IDH2 and control siRNA (left). The number of migrating cells was calculated and analyzed (right). Scale bars, 100 μ m. **e,** A549 cells (left), HCT116 cells (middle), H1299 cells (right) transfected with siRNAs targeting ME2 or control siRNA were lysed and analyzed by western blotting with the indicated antibodies.

Reviewer #2 - PI3K/AKT, biochemistry, phospho proteomics (Remarks to the Author):

In this work, Professor Jiang and colleagues have identified a novel phospho-regulatory mechanism by which growth factor signaling can alter the metabolism of cancer cells. This began with the observation that the metabolic enzyme malic enzyme 2 (ME2), normally mitochondrial, was localized to the cytosol specifically in PTEN null cells. The authors then demonstrated an elegant mechanism to explain this: the growth factor signaling kinase AKT targets and phosphorylates ME2 at its mitochondrial leader sequence (Ser9 – which they identified by mass spec), which blocks its entry into the mitochondrial matrix. Once phosphorylated and trapped in the cytosol, ME2 has higher catalytic activity and functions as a scaffold for multiple glycolytic enzymes, thereby increasing the glycolytic flux to promote anabolic processes and cell growth.

Overall, this solid study reveals a novel and biochemically plausible regulatory mechanism for coordinating growth factor signaling and metabolism. I can imagine this being of interest in cancer and other biological fields as this work involves the phosphoregulation of a widely expressed metabolic enzyme (ME2) by a major signaling kinase (AKT).

We very much thank the referee for considering the topic to be interesting, and appreciate the insightful and constructive comments on our work. As detailed below, we have now provided experimental evidence or detailed explanations of these comments, which we believe significantly improve the manuscript. Please note that new figures generated during the revision are indicated in red color.

Below, I have outlined points for the authors to address.

1. The phosphorylation site (Ser9) on ME2 is arginine-rich and matches the known substrate-targeting motif of AKT. However, it also conforms quite well to S6K's motif (a major downstream kinase of AKT), and thus S6K might cooperate with AKT in regulating the localization of ME2. Can the authors examine the effects of S6K-specific inhibitors on Ser9 phosphorylation in the insulin treatment setting (or alternatively in other related experimental settings)?

We very much thank the referee for this insightful comment on S6K might cooperate with AKT in regulating ME2. To address this comment, we first investigated the effect of S6K-specific inhibitors on ME2c phosphorylation in the context of insulin treatment. Indeed, the referee's speculation is correct. Treating the cells with the specific inhibitors of S6K1, PF4708671 or LY2584702, significantly reduced the phosphorylation of ME2c when the cells were incubated with insulin (**revised Extended Data Fig. 14a**). S6K1 is known to be phosphorylated by mTORC1, which can be triggered by PI3K-AKT signaling activation. Similar results were obtained when cells were treated with the mTORC1 inhibitor rapamycin or torin1. Inhibition of mTORC1 strongly inhibited S6K1 and AKT, resulting in blocked ME2c phosphorylation (**revised Extended Data Fig. 14a**).

In addition, the above findings prompted us to further investigate whether S6K1 itself can bind to and phosphorylate ME2c, as the phosphorylation site (Ser 9) on ME2 also matches the S6K motif quite well. Interestingly, when co-expressed in HEK293 cells, S6K1 formed a complex with

ME2c, regardless of the tag used (**revised Extended Data Fig. 14b, c**). Moreover, like AKT1, introduction S6K1 led to a strong phosphorylation of ME2c (**revised Extended Data Fig. 14d**). Notably, S9A mutation almost completely abolished the phosphorylation of ME2c by S6K1 (**revised Extended Data Fig. 14e**), indicating that Ser 9 on ME2c is also a phosphorylation site of S6K1.

In sum, these findings suggest that S6K1 may indeed act in concert with AKT on ME2c. We agree that further endogenous binding assays are needed to confirm these observations, and given that this may be not the main point of the current work, we hope to return to this in the future for a comprehensive study and wish the reviewer would agree.

2. The authors show that cells expressing the S9E mutant (that is partially retained in the cytosol) are more proliferative. Two underlying mechanisms are proposed to explain this: a) a catalytic-independent mechanism where ME2 promotes the assembly of glycolytic enzymes to enhance glycolytic flux and b) a catalytic-dependent mechanism where phosphorylation enhances its enzymatic activity and NADPH generation in the cytosol (and correspondingly, its absence of activity in the mitochondria and inability to contribute NADH).

Would introducing catalytically inactivating mutations (for example, D279A, G446D, or R670Q (PMID: 26008970)) into ME2 S9E impact its effects on cell proliferation? These questions may be satisfactorily addressed by shorter time frame experiments, such as colony formation assays with ME2 S9E catalytic dead knock-in expression cell lines (ED Fig 10a).

We thank the referee for this excellent comment. To address this comment, we generated a ME2cS9D mutant (ME2cS9D-VGA) that lost enzymatic activity by introducing the VGA mutations (L419G, S420V, N421A)³ (please see **Response Fig. 4a below**). Results from Seahorse XF Extracellular Flux Analysis revealed that VGA mutations minimally affected ME2cS9D-mediated enhancement of glycolysis (please see **Response Fig. 4b below**). These results suggest that the ME2cS9D-VGA mutant could be used for further preliminary functional studies to investigate the relative contribution of enzymatic activity and glycolytic regulation of ME2c to cell proliferation.

Remarkably, in a soft agar assay, expression of ME2cS9D promoted cell proliferation compared to wild-type ME2c and ME2cS9A. Notably, the VGA mutations reduced the pro-proliferative ability of ME2cS9D, as ME2cS9D-VGA-expressing cells proliferated significantly slower than cells expressing ME2cS9D (please see **Response Fig. 4c below**). However, ME2cS9D-VGA-expressing cells still had a comparable level of proliferation to wild-type cells (please see **Response Fig. 4c below**), suggesting that the glycolysis-promoting ability of ME2cS9D does play a role in helping cells to proliferate.

As we are currently unsure whether there are other mechanisms to regulate proliferation besides glycolytic regulation, further studies at the cellular, biochemical and animal model levels are needed to determine the exact contribution of ME2c-regulated glycolysis to cell proliferation. Thus, to fully understand this issue seems to be a long-term project and we hope to address it in a future work and hope that the referee would agree.

Response Fig. 4. Effect of VGA mutations on ME2cS9D-mediated promotion of glycolysis and cell proliferation. **a**, Enzymatic activity of ME2cS9D and ME2cS9D-VGA with catalytically inactivating mutations VGA (L419G, S420V, N421A) purified from HCT116 cells stably expressing pCDH-Flag-ME2cS9D, and pCDH-Flag-ME2cS9D-VGA as indicated. **b**, HCT116 cells stably expressing wild-type ME2c (pCDH-Flag-ME2c), ME2cS9A (pCDH-Flag-ME2cS9A), ME2cS9D (pCDH-Flag-ME2cS9D), ME2cS9D (pCDH-Flag-ME2cS9D-VGA) or vector control (pCDH-Flag Vec) were treated with 25 mM glucose, 1 μM oligomycin and 100 mM 2-DG at the indicated times for ECAR analysis in a Seahorse XFe96 analyzer. **c**, Colony formation assay of HCT116 cells stably expressing wild-type ME2c (pCDH-Flag-ME2c), ME2cS9A (pCDH-Flag-ME2cS9A), ME2cS9D (pCDH-Flag-ME2cS9D), ME2cS9D (pCDH-Flag-ME2cS9D-VGA) or vector control (pCDH-Flag Vec). Numbers of colonies with a diameter above 20 μm were quantified. Data are means ± SD, **P < 0.01; ***P < 0.001; ****P < 0.0001; ns, no significance; two-tailed Student's t test.

3. The leader sequence (first 18 amino acids) on ME2 contains two cysteines (Cys12 and Cys16) that are adjacent to the identified phosphorylation site (Ser9) and appear well conserved. It is interesting to speculate that the cellular redox state might regulate the mitochondrial entry of ME2 through the oxidation of these cysteines, which would introduce negative charge similar to phosphorylated residues. This may be a mechanism for countering redox stress and might be worth commenting on in the discussion section, as it would be in keeping with the impressive reduction of ROS seen in Fig 5i when ME2 is retained in the cytosol.

We are very grateful to the referee for the thoughtful and insightful comments. To address these comments, we generated cell lines stably expressing ME2c carrying C12SC16S or C12DC16D mutations. The C12SC16S mutations were designed to block the oxidation of the two cysteines. The C12DC16D mutations were made to try to mimic the introduction of negative charges after oxidation at both sites. As it turned out, the referee's suspicions are entirely correct. ME2cC12SC16S was found almost exclusively in mitochondria, whereas the C12DC16D mutation almost completely prevented

ME2c from entering mitochondria (**revised Extended Data Fig. 14f**). Thus, these results suggest that the cellular redox state may indeed regulate the mitochondrial entry of ME2 through the oxidation of cysteines 12 and 16.

Oxidation of two cysteines (Cys12 and Cys16) prevents ME2c from entering the mitochondria and, in this case, inhibits the TCA cycle-mediated oxidative scavenging promoted by ME2. Thus, oxidation of these two cysteines has the effect of both counteracting ROS and leading to ROS accumulation due to the inability to enter the mitochondria to promote TCA cycle-mediated ROS elimination. Therefore, although we found that, similar to ME2c, expression of either mutant reduced intracellular ROS levels (**revised Extended Data Fig. 14h**), the regulation of intracellular redox homeostasis by cysteine oxidation at these two sites is complex, and more in-depth biochemical and cellular mechanistic studies are needed to clarify this point.

Again, we greatly appreciate these excellent comments by the referee and have now incorporated and discussed these new data in the revised version.

4. Introduction/background section: This would benefit from a brief background on ME2 and its canonical roles, as it is the paper's primary focus and is less widely known than AKT (which is discussed at length in the intro).

We appreciate this comment by the referee. Malic enzymes (MEs) catalyze the oxidative decarboxylation of malate to generate pyruvate and either NADPH or NADH^{4,5}. In mammalian cells three ME isoforms have been identified: a cytosolic NADP⁺-dependent isoform (ME1), a mitochondrial NAD/P⁺-dependent isoform (ME2), and a mitochondrial NADP⁺-dependent isoform (ME3), of which ME1 and ME2 are the main isoforms³. These enzymes recycle the TCA cycle intermediate malate into the common TCA cycle carbon source pyruvate and thus may have a regulatory role in matching TCA flux to cellular demand for energy, reducing equivalents, and biosynthetic precursors. Notably, ME2 activity is highly elevated in tumor cells and correlates with tumor progression^{3,6-9}, and NADPH generation and redox control capabilities confer potent oncogenic function to ME2^{3,10}. Interestingly, loss of ME2 in pancreatic ductal adenocarcinoma (PDAC) has been shown to require ME3 for metabolic compensation and survival¹¹. Recent studies have also revealed a role for ME2 in metabolically controlling mutant p53 stability and epigenetic programming^{9,12}.

We have now included this information (marked in dark blue) in the Introduction section of the revised version.

5. Figure 1a-In the legend, please clarify what the colors of the arrows depict. (e.g., black arrows represent mitochondrial localization, and red arrows represent cytosolic or non-mitochondrial localization).

Figure 1c – same as 1a regarding white and green arrows.

We thank the referee for this helpful suggestion. As suggested, we have now clarified what the colors of the arrows mean in each of the figure legends.

6. The label 'ME2c' (where c refers to cytoplasmic) is potentially misleading because it implies that the protein is cytosolic when referring to the full-length form. I understand this label was coined to distinguish the full-length form from the cleaved form 'ME2m' that had entered the mitochondria, as illustrated in Fig 2b. However, using 'ME2c' outside these contexts is potentially confusing (e.g., labeling expression constructs as ME2c, showing confocal images of the mitochondrial localization of ME2c). Unless the authors can provide a compelling reason for keeping it, I feel that the 'c' should be dropped from the manuscript, and only 'ME2m' needs to be specified when appropriate.

We are very grateful to the referee for this suggestion and for their patience in explaining it. We initially chose "c" for the following reasons:

1) "ME2c" (where c refers to cytoplasm) does indeed imply that the protein is cytoplasmic. It is probably a bit clearer than using "f" (meaning full length) and also to distinguish it from the cleaved form "ME2m" which enters the mitochondria.

2) Biochemically, the form that localizes in the cytoplasm is the full-length form, in other words, the full-length ME2 is not the full-length form when it enters the mitochondria and becomes ME2m.

3) More importantly, people are used to calling the mitochondrial form ME2 and renaming it ME2m may cause confusion for some time. So we decided to keep the mitochondrial form as ME2 and just naming the cytoplasmic form ME2.

We are very grateful to the referee for raising the issue that labelling the expression construct as ME2c and the confocal images showing the mitochondrial localization of ME2c might have caused confusion. The DNA in the constructed expression plasmid does indeed use the full-length ME2 sequence (the sequence of ME2c), whereas the mitochondrial localization is no longer full-length and is therefore not considered to be ME2c. We have revised the figure labeling in the revised version to try to be as non-confusing as possible.

Certainly, if the referee still prefers, we are happy to remove the "c" from ME2c in the manuscript and indicate "ME2m" where appropriate. Considering that it may be an extra effort for the other two referees to review the manuscript and pay attention to the change in the name of ME2, we keep the previous naming style during this round of revision and will change the name of ME2c according to this referee's suggestion after the revision is completed, and we sincerely hope that the referee would agree to this.

Reviewer #3 - Prostate cancer, metabolism (Remarks to the Author):

The paper presents an intriguing discovery of a previously unknown mechanism by which AKT1 regulates glycolysis and the mitochondrial TCA cycle through ME2 phosphorylation. The authors provide a detailed account of the interaction between ME2c and key glycolytic enzymes, enhancing our understanding of how tumor cells coordinate glycolysis in response to growth stimuli.

The authors have employed a variety of experimental techniques, including immunoprecipitation analysis, confocal microscopy, and in vivo tumor models, to explore the role of ME2c phosphorylation in tumorigenesis. A more detailed description of the experimental methods and controls used would be helpful to allow readers to better evaluate the validity of the findings.

We are very grateful to the referee for considering our work interesting and appreciate the positive comments. We have now revised the paper, including the experimental methods, according to the referee's suggestions, which we believe have greatly improved our manuscript. Changes and additions in the text are highlighted in dark blue.

Comments:

1. The authors should discuss potential limitations of the study and any assumptions made in interpreting the data. This would provide a more balanced view of the results and their implications.

We very much thank the referee for this important comment. In this study, we found that growth factor stimulation induces AKT-mediated phosphorylation of ME2c, a form of ME2 that is virtually undetectable in the absence of stimulation (which is probably why we have not found this cytoplasmic form to exist to date). The expression of ME2c appears to be low compared to mitochondrial ME2, the accumulation of which leads to an increase in glycolytic activity, causing a Warburg effect in cancer cells. In addition to its tumor-promoting effects, it may also play a role in some proliferating cells (e.g. hematopoietic stem cells, etc.), as these rapidly proliferating cells also tend to be highly glycolytic. Thus, it would be of great interest to further elucidate what physiological functions this newly discovered form of cytoplasmic ME2 has in addition to its tumor-promoting effects in PTEN-deficient or AKT-activated tumors. In this case, genetic modification of ME2c and mitochondrial ME2 expression would be helpful. We have now included this information in the discussion section of the revised version.

It is also interesting to study how exactly ME2c plays a role in metabolic regulation, which we did not fully reveal in this paper. For example, with the help of the other referees, we have provided additional experimental data and discussion on the mechanism of post-translational modification and the regulatory role of ME2c in redox homeostasis.

2. The discussion section could be expanded to include a broader perspective on the role of AKT1 in metabolic reprogramming and its implications for cancer biology. Additionally, the authors should

consider possible future research directions, including the investigation of other molecular mechanisms that may be involved in ME2c phosphorylation and glycolytic regulation.

We are very grateful to the reviewer for these helpful comments. As suggested, we have expanded the discussion to include some information on the role of AKT1 in metabolic reprogramming and cancer cell biology. We have also included discussions of some possible future research directions, including investigating the physiological and glycolytic functions of ME2c and how ME2c is post-translationally modified and metabolically regulated.

In the conclusion, the authors should provide a clear summary of the main findings, emphasizing the novelty of their work and its potential implications for the understanding of tumor cell metabolism and treatment strategies. Better draw an illustrative picture to show molecular mechanisms by end.

We appreciate this excellent comment by the referee. As suggested, we have now summarized the main findings in the discussion section and provided a working model (**revised Extended Data Fig. 15**).

References

1. Hamidi, A., Song, J., Thakur, N., Itoh, S., Marcusson, A., Bergh, A., Heldin, C.H., and Landstrom, M. (2017). TGF-beta promotes PI3K-AKT signaling and prostate cancer cell migration through the TRAF6-mediated ubiquitylation of p85alpha. *Sci Signal* *10*. 10.1126/scisignal.aal4186.
2. Kato, M., Putta, S., Wang, M., Yuan, H., Lanting, L., Nair, I., Gunn, A., Nakagawa, Y., Shimano, H., Todorov, I., et al. (2009). TGF-beta activates Akt kinase through a microRNA-dependent amplifying circuit targeting PTEN. *Nat Cell Biol* *11*, 881-889. 10.1038/ncb1897.
3. Jiang, P., Du, W., Mancuso, A., Wellen, K.E., and Yang, X. (2013). Reciprocal regulation of p53 and malic enzymes modulates metabolism and senescence. *Nature* *493*, 689-693. 10.1038/nature11776.
4. Hsu, R.Y. (1982). Pigeon liver malic enzyme. *Mol Cell Biochem* *43*, 3-26.
5. Chang, G.G., and Tong, L. (2003). Structure and function of malic enzymes, a new class of oxidative decarboxylases. *Biochemistry* *42*, 12721-12733. 10.1021/bi035251+.
6. Sauer, L.A., Dauchy, R.T., Nagel, W.O., and Morris, H.P. (1980). Mitochondrial malic enzymes. Mitochondrial NAD(P)⁺-dependent malic enzyme activity and malate-dependent pyruvate formation are progression-linked in Morris hepatomas. *J Biol Chem* *255*, 3844-3848.
7. Nagel, W.O., Dauchy, R.T., and Sauer, L.A. (1980). Mitochondrial malic enzymes. An association between NAD(P)⁺-dependent malic enzyme and cell renewal in Sprague-Dawley rat tissues. *J Biol Chem* *255*, 3849-3854.
8. Wasilenko, W.J., and Marchok, A.C. (1985). Malic enzyme and malate dehydrogenase activities in rat tracheal epithelial cells during the progression of neoplasia. *Cancer Lett* *28*, 35-42.
9. Zhao, M., Yao, P., Mao, Y., Wu, J., Wang, W., Geng, C., Cheng, J., Du, W., and Jiang, P. (2022). Malic enzyme 2 maintains protein stability of mutant p53 through 2-hydroxyglutarate. *Nat Metab* *4*, 225-238. 10.1038/s42255-022-00532-w.
10. Li, W., Kou, J., Zhang, Z., Li, H., Li, L., and Du, W. (2023). Cellular redox homeostasis maintained by malic enzyme 2 is essential for MYC-driven T cell lymphomagenesis. *Proc Natl Acad Sci U S A* *120*, e2217869120. 10.1073/pnas.2217869120.
11. Dey, P., Baddour, J., Muller, F., Wu, C.C., Wang, H., Liao, W.T., Lan, Z., Chen, A., Gutschner, T., Kang, Y., et al. (2017). Genomic deletion of malic enzyme 2 confers collateral lethality in pancreatic cancer. *Nature* *542*, 119-123. 10.1038/nature21052.
12. Li, W., Kou, J., Qin, J., Li, L., Zhang, Z., Pan, Y., Xue, Y., and Du, W. (2021). NADPH levels affect cellular epigenetic state by inhibiting HDAC3-Ncor complex. *Nat Metab* *3*, 75-89. 10.1038/s42255-020-00330-2.

Reviewers' Comments:

Reviewer #2:

Remarks to the Author:

The authors have addressed my concerns in full.

As stated, I believe this will be a valuable paper to the field.

Reviewer #3:

Remarks to the Author:

accept after language refinement.

Reviewer #4:

Remarks to the Author:

The additional data in this revised manuscript has satisfactorily addressed the concerns raised by Reviewer 1.

I have some additional comments:

1. In general the biochemistry characterization of ME2-c (should be more appropriately termed full length Me2, more on that later) and its interaction with glycolytic enzymes appeared well executed and convincing. However, the imaging data provided to support the cytosolic localization of ME2-c appeared to be of poor quality and not convincing. Some of the areas in the EM micrographs in Fig 1a used dotted lines to mark areas purportedly showing mitochondria. However, most of these regions failed to show typical mitochondrial morphology such as double membranes and cristae structures. It is also curious to note that mitochondria in IgG control in Fig. 1a has more typical mitochondria morphology (with some cristae structure and double membrane structure) than in ME2 staining micrograph. This is really unfortunate, since one is unable to determine whether these immunogold dots are inside mitochondria or not. These data are clearly not of publication qualities and it is hard for one to determine the localization of ME2 in these cells.
2. The immunofluorescence staining of ME2 and mitochondria (with Mitotracker Red) in many figure panels was of poor quality, especially for figure panels in extended figures. For example, in Extended Fig. 2A Mitotracker staining showed blurry and smear red staining without showing distinct beads or threads like mitochondria morphology. It seemed important for authors to provide higher quality immunofluorescent staining data to support the conclusion that Akt1 phosphorylation and PTEN inactivation promote the cytosolic localization of ME2.
3. I agree with review 2's comment that labeling full length ME2 as ME2-c is potentially misleading and should be corrected. Most mitochondrial matrix protein (like ME2) or mitochondrial inner membrane protein has a positively charged mitochondrial localization signaling peptide at the N-terminal. When imported into the mitochondria the signaling peptide is cleaved to produce mature mitochondrial protein. The way it is described in this manuscript, it appeared that this is an unknown new variant of ME2, which is misleading. I found the last sentence on page 15 " Thus, the previously widely recognized ME2 protein is actually a truncated form of ME2c protein with the N-terminal signal peptide missing." especially misleading. Cleavage of the N-terminal signaling peptide after mitochondrial import is a universal mechanism for the processing of mitochondrial matrix protein and proteins targeted to other organelles, not something unique to ME2.
4. The statement/conclusion that C12/C16 oxidation could potentially regulate mitochondrial localization of ME2 is not convincing (Extended Fig 14g). The C12S/C16S mutant has similar cytosol to mitochondria ratios as WT. Although C to D mutation did increase cytosolic localization, the same effect could probably achieved by mutating any residues to D in the N-terminal signal peptide region.

Point-by-point response to Reviewers' comments

Reviewer #2 (Remarks to the Author):

The authors have addressed my concerns in full.

As stated, I believe this will be a valuable paper to the field.

Reviewer #3 (Remarks to the Author):

accept after language refinement.

We appreciate the constructive comments by all the referees.

Reviewer #4 (Remarks to the Author):

The additional data in this revised manuscript has satisfactorily addressed the concerns raised by Reviewer 1.

We thank the referee for considering that we have satisfactorily addressed the concerns raised by reviewer #1.

I have some additional comments:

1. In general the biochemistry characterization of ME2-c (should be more appropriately termed full length Me2, more on that later) and its interaction with glycolytic enzymes appeared well executed and convincing. However, the imaging data provided to support the cytosolic localization of ME2-c appeared to be of poor quality and not convincing. Some of the areas in the EM micrographs in Fig 1a used dotted lines to mark areas purportedly showing mitochondria. However, most of these regions failed to show typical mitochondrial morphology such as double membranes and cristae structures. It is also curious to note that mitochondria in IgG control in Fig. 1a has more typical mitochondria morphology (with some cristae structure and double membrane structure) than in ME2 staining micrograph. This is really unfortunate, since one is unable to determine whether these immunogold dots are inside mitochondria or not. These data are clearly not of publication qualities and it is hard for one to determine the localization of ME2 in these cells.

We thank the referee for this comment. Indeed, there are several reasons why ideal images could not be obtained, for instance, staining of mitochondria usually affects the sensitivity of the antibody, and it is difficult for the microscope to focus on both the antibody-stained particles (immunogold dots) and the mitochondria at the same time (there may be a difference in focal length). Nevertheless, we have now replaced the EM micrographs in Figure 1a with higher resolution micrographs that clearly show differences in the localization of ME2 inside and outside the mitochondria.

2. The immunofluorescence staining of ME2 and mitochondria (with Mitotracker Red)

in many figure panels was of poor quality, especially for figure panels in extended figures. For example, in Extended Fig. 2A Mitotracker staining showed blurry and smear red staining without showing distinct beads or threads like mitochondria morphology. It seemed important for authors to provide higher quality immunofluorescent staining data to support the conclusion that Akt1 phosphorylation and PTEN inactivation promote the cytosolic localization of ME2.

As suggested, we have repeated these experiments and provided higher quality immunofluorescence staining data.

3. I agree with review 2's comment that labeling full length ME2 as ME2-c is potentially misleading and should be corrected. Most mitochondrial matrix protein (like ME2) or mitochondrial inner membrane protein has a positively charged mitochondrial localization signaling peptide at the N-terminal. When imported into the mitochondria the signaling peptide is cleaved to produce mature mitochondrial protein. The way it is described in this manuscript, it appeared that this is an unknown new variant of ME2, which is misleading. I found the last sentence on page 15 " Thus, the previously widely recognized ME2 protein is actually a truncated form of ME2c protein with the N-terminal signal peptide missing." especially misleading. Cleavage of the N-terminal signaling peptide after mitochondrial import is a universal mechanism for the processing of mitochondrial matrix protein and proteins targeted to other organelles, not something unique to ME2.

As suggested, ME2c has now been changed to ME2fl (fl denotes full length) throughout the manuscript.

4. The statement/conclusion that C12/C16 oxidation could potentially regulate mitochondrial localization of ME2 is not convincing (Extended Fig 14g). The C12S/C16S mutant has similar cytosol to mitochondria ratios as WT. Although C to D mutation did increase cytosolic localization, the same effect could probably achieved by mutating any residues to D in the N-terminal signal peptide region.

We thank the referee for this comment, and agree this statement need to be revised. Therefore, we have toned down the statement to "oxidation of these cysteines might affect the subcellular localization of ME2fl....." on page 16.